# Gene duplication and deletion caused by over-replication at a fork barrier

Judith Oehler [1,2], Carl A. Morrow [1,2] & Matthew C. Whitby [1] ✉

Replication fork stalling can provoke fork reversal to form a four-way DNA junction. This remodelling of the replication fork can facilitate repair, aid bypass of DNA lesions, and enable replication restart, but may also pose a risk of over-replication during fork convergence. We show that replication fork stalling at a site-specific barrier in fission yeast can induce gene duplication-deletion rearrangements that are independent of replication restart-associated template switching and Rad51-dependent multi-invasion. Instead, they resemble targeted gene replacements (TGRs), requiring the DNA annealing activity of Rad52, the 3'-flap nuclease Rad16-Swi10, and mismatch repair protein Msh2. We propose that excess DNA, generated during the merging of a canonical fork with a reversed fork, can be liberated by a nuclease and integrated at an ectopic site via a TGR-like mechanism. This highlights how over-replication at replication termination sites can threaten genome stability in eukaryotes.

The reshaping of genomes through the gain and loss of genetic material is a potent driver of evolution and disease. For example, segmental duplications, which are >1 kbp genomic regions that have been copied and re-introduced at a new chromosomal site, have played a key role in primate evolution[1]. And templated DNA insertions and DNA deletions are common features of cancer genomes that can drive tumour development by altering the expression of tumour-suppressor genes and oncogenes through the amplification, loss, disruption and fusion of different coding and regulatory segments[2]. Many genomic duplications and deletions stem from mistakes made during the normal processes of DNA replication and repair. In particular, the inadvertent recombination of non-allelic/ectopic DNA sequences, during the repair of DNA double-strand breaks (DSBs) by homologous recombination (HR) and non-homologous end-joining (NHEJ), plays a well-documented role in generating copy number variants[3]. Similarly, the repair of collapsed/broken replication forks by break-induced replication (BIR), or the related process of microhomology-mediated BIR, is thought to account for many of the genomic rearrangements, including DNA deletions and insertions, that are observed in healthy and morbid genomes[3–5].

One aspect of DNA metabolism whose potential for generating genomic duplications and deletions is less well established is the process of DNA replication termination where replication forks merge. This is an event that happens tens of thousands of times in each human cell cycle and, therefore, must normally be a seamless and non-mutagenic process[6,7]. However, it is hypothesised that some termination events may be liable to generate regions of over-replicated DNA which, if not degraded, could be inserted at new chromosomal sites[8,9]. Evidence for this idea comes from a study in budding yeast that revealed high rates of templated DNA insertions at DSBs in cells lacking the DNA replication helicase/nuclease Dna2[8]. Importantly, many of the DNA insertions were from regions of the yeast genome where replication fork stalling is frequent. Stalled replication forks can reverse into a four-way DNA junction or "chicken foot" structure and, therefore, replication termination may sometimes occur by the convergence of a canonical replication fork with a reversed fork, especially under conditions of replication stress where fork stalling and reversal are more common[9–11]. This could result in over-replication of the DNA spanning the region through which the stalled fork had reversed. Dna2 has been implicated in processing reversed replication forks where it is thought to degrade the regressed arm of the chicken foot structure[12,13]. Therefore, in its absence, the over-replicated DNA at terminating reversed forks could persist and end up being excised and incorporated at a new genomic site[8]. However, direct evidence that fork

[1]Department of Biochemistry, University of Oxford, South Parks Road, Oxford OX1 3QU, UK. [2]These authors contributed equally: Judith Oehler, Carl A. Morrow. ✉e-mail: matthew.whitby@bioch.ox.ac.uk

stalling can result in over-replication and release of a DNA fragment that can be integrated at a nearby or distant genomic site is lacking.

In this study, we provide direct evidence that over-replication of DNA can occur at sites of replication fork stalling and reveal how this can lead to templated DNA insertions and deletions even in the presence of Dna2. We also identify key factors that promote DNA insertions and those that limit their occurrence.

## Results

### Experimental system for measuring duplication-deletion rearrangements

If over-replication occurs when a canonical replication fork converges with a reversed fork, then the excess DNA, if not efficiently degraded, could be excised and integrated at an ectopic site by HR (Supplementary Fig. 1a). We will refer to this model as Duplication by Reversed Fork Termination (DRFT). To investigate whether DRFT happens in vivo, we developed a genetic assay in the fission yeast *Schizosaccharomyces pombe* incorporating the strong polar replication fork barrier (RFB) *RTS1*, a hygromycin resistance gene (*hyg^R*), and a *ura4* gene required for uracil biosynthesis (Fig. 1a). When a replication fork encounters *RTS1* in its blocking orientation, it stalls and reverses but remains unbroken[14–16]. The reversed fork is then resolved either by a converging fork or by HR proteins driving its restart through a process termed recombination-dependent replication (RDR)[17–19]. RDR commences within ~18 min of the fork encountering the barrier and, similar to BIR, exhibits features that distinguish it from canonical DNA replication including a propensity to undergo template switching[17,20–22].

By siting *RTS1* at a location on chromosome 3 where DNA replication is essentially unidirectional (moving from left to right across the barrier as depicted in Fig. 1a), we could define the orientation of the RFB that would block replication forks (we refer to this as the Active Orientation or AO), as well as the direction of fork reversal. Fork blocking at *RTS1*-AO was confirmed by analysing replication intermediates at the barrier by native two-dimensional gel electrophoresis (2DGE) (Fig. 1b). We also confirmed that when *RTS1* is switched to its Inactive Orientation (IO), replication forks moving in the left to right direction pass through the barrier relatively unhindered (Fig. 1b). We will refer to the region to the right of *RTS1* as upstream and the region to its left as downstream. *Hyg^R* is located immediately upstream of *RTS1* so is the sequence that is predicted to be over-replicated in our model (Supplementary Fig. 1a). It is flanked by H2b and H3b that encompass the promoter and terminator sequences of the *TEF* gene from *Ashbya gossypii* (Fig. 1a). Identical or almost identical sequences (H2a and H3a) flank the *ura4* gene that is sited ~11.4 kb downstream of *RTS1*. Therefore, if fork stalling at *RTS1* causes DRFT, then *hyg^R* together with its flanking H2b and H3b sequences will be over-replicated, excised and integrated at the *ura4* site resulting in a duplication of *hyg^R* and deletion of *ura4*. This type of genomic rearrangement is termed a duplication-deletion (Dup-Del) and is measured in our genetic assay by determining the frequency of Ura- colonies through selection on media containing 5-Fluoroorotic Acid (FOA) and then confirming the Dup-Del rearrangement by two PCR amplifications using primer sets A and B (Fig. 1a).

### RTS1-AO strongly induces Dup-Dels

With *RTS1*-AO there is a ~1488-fold increase in the frequency of Ura- colonies compared to the strain containing *RTS1*-IO (Fig. 1c). PCR analysis of Ura- colonies confirmed that the vast majority contained the Dup-Del rearrangement (Fig. 1d). However, a few of the Ura- colonies in the *RTS1*-IO strain contained the starting configuration of *hyg^R* and *ura4* and were assumed to have acquired a loss-of-function mutation in either *ura4* or *ura5*[23]. There were also some colonies whose Dup-Del status remained undefined due to a failure of one or both PCRs (see Methods). When these factors are taken into account, we can

estimate that the increase in Dup-Del formation in the *RTS1*-AO strain is ~1716-fold compared to the *RTS1*-IO strain (Supplementary Data 1).

To further validate the Dup-Del rearrangement, we analysed purified genomic DNA from a random selection of Ura- colonies by both PCR and Southern blot analysis (Fig. 1e, f and Supplementary Figs. 2 and 3). Unlike the colony PCR analysis, where some PCRs failed to generate a DNA band, all of the PCRs done with purified genomic DNA yielded a band. This suggests that cases of undefined Dup-Del status by colony PCR analysis arise mainly because of a technical problem with the PCR rather than some undetermined genomic rearrangement that cannot be amplified with primer sets A or B. Out of 40 Ura- colonies tested by PCR of genomic DNA from the *RTS1*-AO strain, all were confirmed to contain a Dup-Del (Fig. 1e and Supplementary Fig. 2). A similar result was obtained for the *RTS1*-IO strain where 32 out of 37 Ura- colonies exhibited a Dup-Del. However, in this strain background, we also identified 5 colonies that displayed a presumed loss-of-function mutation in *ura4/ura5* (Fig. 1e and Supplementary Fig. 2). Intriguingly, Southern blot analysis revealed the presence of three classes of Dup-Del: the expected class, consisting of a duplication of *hyg^R* and deletion of *ura4* (Dup-Del 1), which accounts for the majority of Dup-Dels (36/40 *RTS1*-AO, and 20/32 *RTS1*-IO); and two minor classes (Dup-Del 2 and Dup-Del 3) where a *hyg^R* plus *RTS1* fragment, flanked by H1b and H3b, is duplicated and integrated at the downstream homologous sites H1a/H1c and H3a (Fig. 1a). We suspect that Dup-Dels 2 and 3 arise from DRFT following extension of the over-replicated DNA beyond *RTS1* by RDR.

### Neither RDR-associated template switching nor Rad51-mediated multi-invasion are required for Dup-Del formation

There are two alternative models to DRFT that could account for Dup-Del formation: 1) RDR-associated template switching (Supplementary Fig. 1b)[17,19,22]; and 2) multi-invasion from a reversed replication fork (Supplementary Fig. 1c)[24]. In the first model, Dup-Del formation could occur if RDR progresses to the H3a sequence adjacent to *ura4* and then undergoes a template switch event that relocates the elongating DNA strand to the H3b sequence next to *hyg^R*. RDR could then copy the *hyg^R* gene before undergoing a second template switch event, shifting the elongating strand from the H2b sequence to the H2a sequence next to *ura4*. In the second model, the loading of Rad51 onto the reversed replication fork could result in strand invasion of the H2a and H3a sequences flanking *ura4* by the H2b and H3b sequences flanking *hyg^R*. This type of multi-invasion could potentially result in a Dup-Del (Supplementary Fig. 1c). To determine whether Dup-Del formation is driven by either RDR-associated template switching or multi-invasion, we investigated whether it shares the same requirements as these mechanisms.

In previous work, we showed that delaying fork convergence by deleting the strong centromere-proximal replication origin, *ori-1253*, provides more time for the collapsed fork at *RTS1* to recruit recombination proteins, be restarted, and progress via RDR towards a downstream *ade6-* direct repeat reporter[17,19,22]. Consequently, this delay results in an increase in Ade+ recombinants due to template switching occurring between the two *ade6-* heteroalleles[17,19,22]. To enable a direct comparison between RDR-associated template switching and Dup-Del formation, we positioned the *ade6-* direct repeat reporter adjacent to *ura4* (Fig. 2a). This configuration allows us to simultaneously assess whether Ade+ recombinants and Dup-Dels increase when *ori-1253* is deleted, which, if observed, would suggest that they are both generated through RDR-associated template switching. As expected, deletion of *ori-1253* resulted in a ~6-fold increase in *RTS1*-AO-induced Ade+ recombinants at the *ade6-* direct repeat reporter (Fig. 2b)[17,19,22]. In contrast, the presence or absence of *ori-1253* had no effect on Dup-Del frequency indicating that RDR-associated template switching is not required for their formation (Fig. 2c, d). We also determined the frequency of Ade+ recombinants

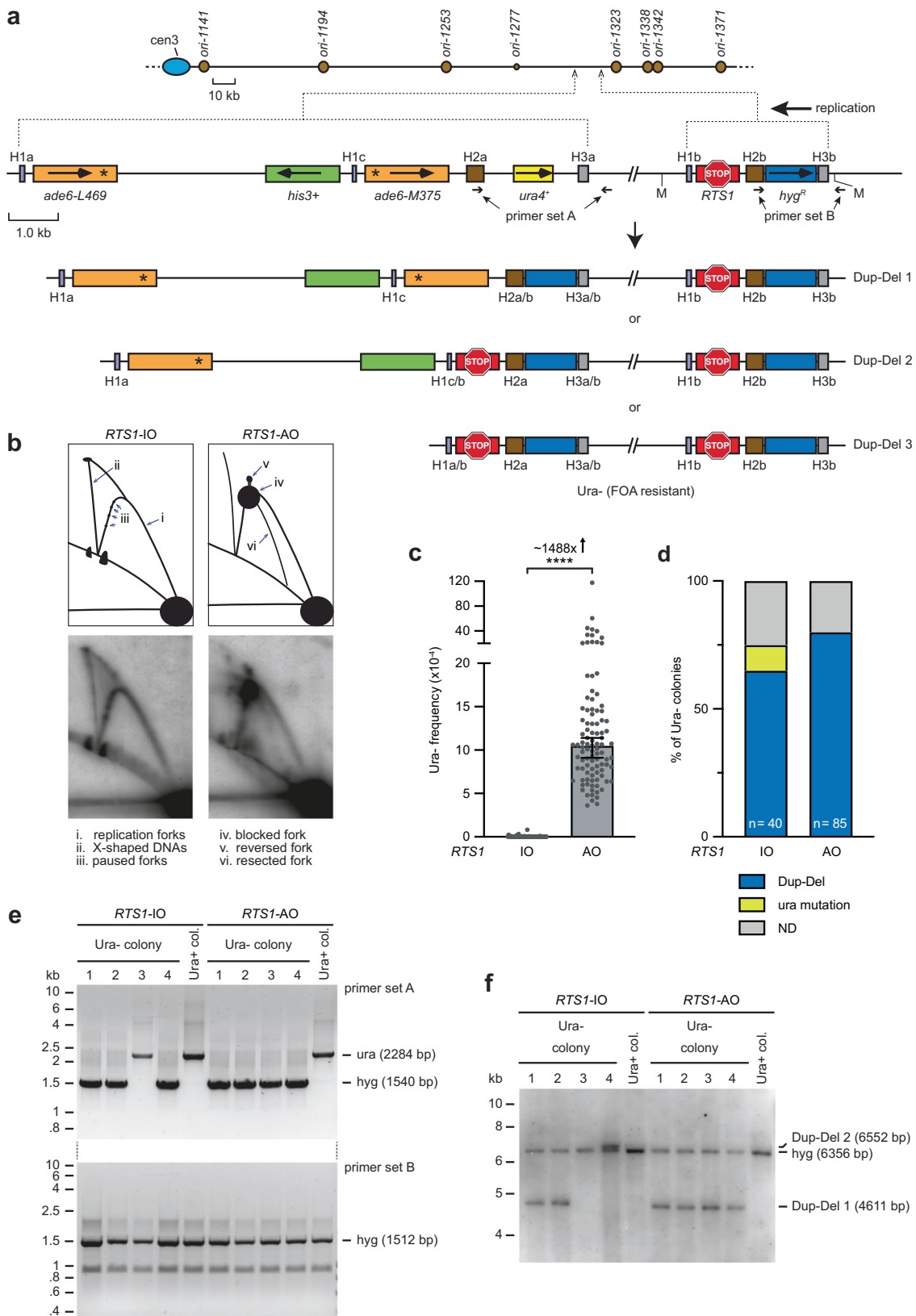

i. replication forks
ii. X-shaped DNAs
iii. paused forks

iv. blocked fork
v. reversed fork
vi. resected fork

amongst FOA-resistant colonies to see if Dup-Del formation and RDR-associated template switching are linked (Supplementary Table 1). We observed a similar frequency of Ade+ recombinants amongst FOA-resistant colonies as in the total cell population implying that Dup-Del formation and RDR-associated template switching at the *ade6⁻* direct repeat reporter are independent processes.

We next assessed the effect of placing the *ade6⁻* direct repeat reporter and adjacent *ura4* gene at sites more distal to *RTS1* (Fig. 3a). As shown previously, when *ori-1253* is present, *RTS1*-AO induced Ade+ recombinants can be detected up to 35 kb downstream of the barrier but not at 75 kb and 140 kb downstream (Fig. 3b)[22]. In contrast, *RTS1*-AO-induced Dup-Dels can be detected at all three sites, albeit their

**Fig. 1 | Replication fork stalling at *RTS1* promotes Dup-Del formation. a** Diagram of the Dup-Del reporter showing its location on chromosome 3 and the three classes of Dup-Del. The marker genes are indicated by the orange, green, yellow and blue rectangles with the arrows indicating the direction of transcription. Replication origins (brown circles), *RTS1* RFB (red stop symbol), relevant MfeI restriction sites (M), and primer sites are also indicated. Information about the H1a/b/c (purple boxes), H2a/b (brown boxes), and H3a/b (grey boxes) sequences is given in Supplementary Figures 3 and 5. **b** Native 2DGE analysis of replication intermediates. Top: schematic of the replication intermediates in the MfeI restriction fragment encompassing *RTS1* shown in **a** detected using *hyg^R* as the probe. Bottom: representative 2D gel images from three independent experiments. **c** Frequency of spontaneous (*RTS1*-IO) and *RTS1*-AO-induced Ura- (FOA resistant) colonies in wild-type strains carrying the Dup-Del reporter shown in Panel **a**. Data are presented as median values ± 95% confidence interval with individual data points shown as grey dots. *P* value = < 0.0001 (****) calculated by the Mann-Whitney test (two-tailed). The

data are also reported in Supplementary Data 1, which includes the strain numbers, the number of colonies tested for each strain (*n*) and *p* value. Further details of the statistical analysis are reported in Supplementary Data 2. **d** Percentage of Ura-colonies containing a Dup-Del or putative mutation in *ura4/ura5* determined by colony PCR. ND indicates a failure of one or both diagnostic PCRs, and n indicates the number of independent Ura- colonies tested. **e** PCR analysis of genomic DNA purified from independent Ura+ and Ura- colonies derived from wild-type strains carrying the Dup-Del reporter shown in Panel **a**. PCR analysis of additional Ura+ and Ura- colonies is shown in Supplementary Fig. 2. **f** Southern blot analysis of genomic DNA extracted from the same colonies analysed in Panel **e**. The DNA was cut with AflII and EcoNI, and restriction fragments detected using *hyg^R* as the probe. The restriction map is shown in Supplementary Fig. 3a. Southern blot analysis of additional Ura+ and Ura- colonies is shown in Supplementary Fig. 3b and c. Source data are provided as a Source Data file.

frequency declines the further *ura4* is from *RTS1* (Fig. 3c, d). We also tested whether Dup-Dels can be detected when the *ura4* gene is on a different chromosome (chr. II) to *RTS1* (Fig. 3a). As expected, there was no difference in the frequency of Ade+ recombinants between the *RTS1*-IO and *RTS1*-AO strains (Fig. 3b). There was also no difference in the frequency of Ura- colonies (Fig. 3c). Crucially, however, PCR analysis revealed that the majority of Ura- colonies from the *RTS1*-AO strain contained Dup-Dels whereas those from the *RTS1*-IO strain contained only Ura- mutations (Fig. 3d). These data provide further evidence that RDR-associated template switching is not the main driver of *RTS1*-AO-induced Dup-Dels.

A key feature of multi-invasion recombination is a dependence on the Rad51 recombinase[24]. However, the estimated frequency of *RTS1*-AO-induced Dup-Dels increases by ~3.6-fold in a *rad51Δ* mutant suggesting that multi-invasion is not required for their formation (Fig. 4 and Supplementary Data 1). The increase in Dup-Del formation also contrasts with RDR-associated template switching, which exhibits either a modest reduction or no change in a *rad51Δ* mutant (Supplementary Fig. 4)[19].

## Dup-Del formation has similar genetic requirements as targeted gene replacement

Our DRFT model predicts that the over-replicated DNA fragment is integrated at an ectopic site via a reaction akin to targeted gene replacement (TGR)[25,26]. Although an analysis of the genetic requirements of TGR has not been performed in *S. pombe*, we assume that they will be similar to those in the budding yeast *Saccharomyces cerevisiae*. Amongst the factors required for efficient TGR in this organism are Rad51, Rad52, the Rad1-Rad10 heterodimeric nuclease (*S. pombe* Rad16-Swi10), and the mismatch repair protein Msh2[27–32]. Although Dup-Dels increase in a *rad51Δ* mutant, they are abolished in a *rad51Δ rad52Δ* double mutant indicating that HR is required for their formation (Fig. 4). Importantly, mutation of conserved arginine-45 in Rad52, which disables its DNA annealing activity but not its role as a mediator for Rad51 DNA loading[19,33], reduces the frequency of *RTS1*-AO-induced Ura- colonies by ~371-fold with an estimated reduction in Dup-Dels of ~641-fold (Fig. 4 and Supplementary Data 1). And deletion of *rad16/swi10/msh2* results in a ~50-fold reduction in Dup-Dels, whilst having little or no effect on the frequency of Ade+ recombinants formed by template switching (Fig. 4, Supplementary Fig. 4, and Supplementary Data 1). Altogether these data show that *RTS1*-AO induced Dup-Del formation has similar genetic requirements as TGR in *S. cerevisiae*, which is consistent with our proposed DRFT model. One exception is Rad51 that, despite being required for TGR in *S. cerevisiae*, suppresses Dup-Del formation. Interestingly, whilst the frequency of TGR in *S. cerevisiae* is reduced by ~1000-fold in a *rad52Δ* mutant, it is only reduced by ~8-fold in a *rad51Δ* mutant indicating that TGR is not completely dependent on Rad51[30]. We suspect that the increase in Dup-Dels observed in a *rad51Δ* mutant reflects an inhibitory effect of

Rad51 prior to TGR. We also think that this increase compensates for any reduction in TGR that a *rad51Δ* mutant might exhibit.

## Long-range DNA end resection is required for efficient Dup-Del formation

The reversed replication fork at *RTS1* undergoes a two-step resection process to generate a single-strand (ss) DNA tail, which is subsequently bound by Rad52 and Rad51. The initial phase involves short-range resection performed by the Mre11-Rad50-Nbs1 complex in conjunction with Ctp1. Following this, a more extensive long-range resection is catalysed by Exonuclease 1 (Exo1)[16,34,35]. Interestingly, the alternative long-range resection pathway in eukaryotes, which involves a RecQ-type helicase (Rqh1 in fission yeast) in cooperation with Dna2, is not implicated in processing forks stalled at *RTS1*[16].

To investigate whether long-range resection is required for Dup-Del formation, we initially examined an *exo1Δ* mutant. Previous research had indicated that long-range resection is not required for RDR[16], and in line with this, we observed no reduction in template switching when *exo1* was deleted (Supplementary Data 1). However, we did observe a modest reduction in Dup-Del frequency, although this reduction did not reach statistical significance (Fig. 5 and Supplementary Data 1). Although the Rqh1-Dna2 long-range resection pathway is not implicated in processing stalled forks at *RTS1*, we tested a *rqh1Δ exo1Δ* double mutant (Fig. 5 and Supplementary Data 1). While there was no significant change in the frequency of template switching, surprisingly the double mutant exhibited a ~4-fold reduction in Dup-Del formation. This result suggests that efficient Dup-Del formation is contingent upon long-range DNA end resection.

## Mismatch repair proteins can abort Dup-Del formation

Msh2, functioning as a heterodimer with Msh3, binds to insertion/deletion loops (IDLs) in DNA and is thought to promote TGR by aiding the processing of recombination intermediates formed at the junction between homologous and heterologous DNA sequences by Rad1-Rad10[32]. Msh2 also forms a heterodimer with Msh6, which is key for the recognition of base-base mismatches and small IDLs during mismatch repair[36]. Unlike Msh2, Msh6 is not required for TGR and accordingly we observed no reduction in Dup-Del formation in a *msh6Δ* mutant (Fig. 6 and Supplementary Data 1)[32]. Instead, a *msh6Δ* mutant exhibited a > 5-fold increase in *RTS1*-AO-induced Dup-Dels. The homologous sequences flanking *hyg^R* and *ura4* contain a single base-base mismatch and 3 nucleotide IDL at the telomere proximal end of H3a and H3b (Supplementary Fig. 5). These mismatches could be recognised by Msh2-Msh3/Msh6 during TGR leading to rejection of the over-replicated *hyg^R* containing DNA fragment[37]. Surprisingly, deletion of the MutL homologue Mlh1 caused the same increase in Dup-Del formation as a *msh6Δ* mutant (Fig. 6 and Supplementary Data 1). Previous studies have reported a lesser role for Mlh1 in heteroduplex rejection during HR than Msh2-Msh3/Msh6[37]. However, for aborting Dup-Del

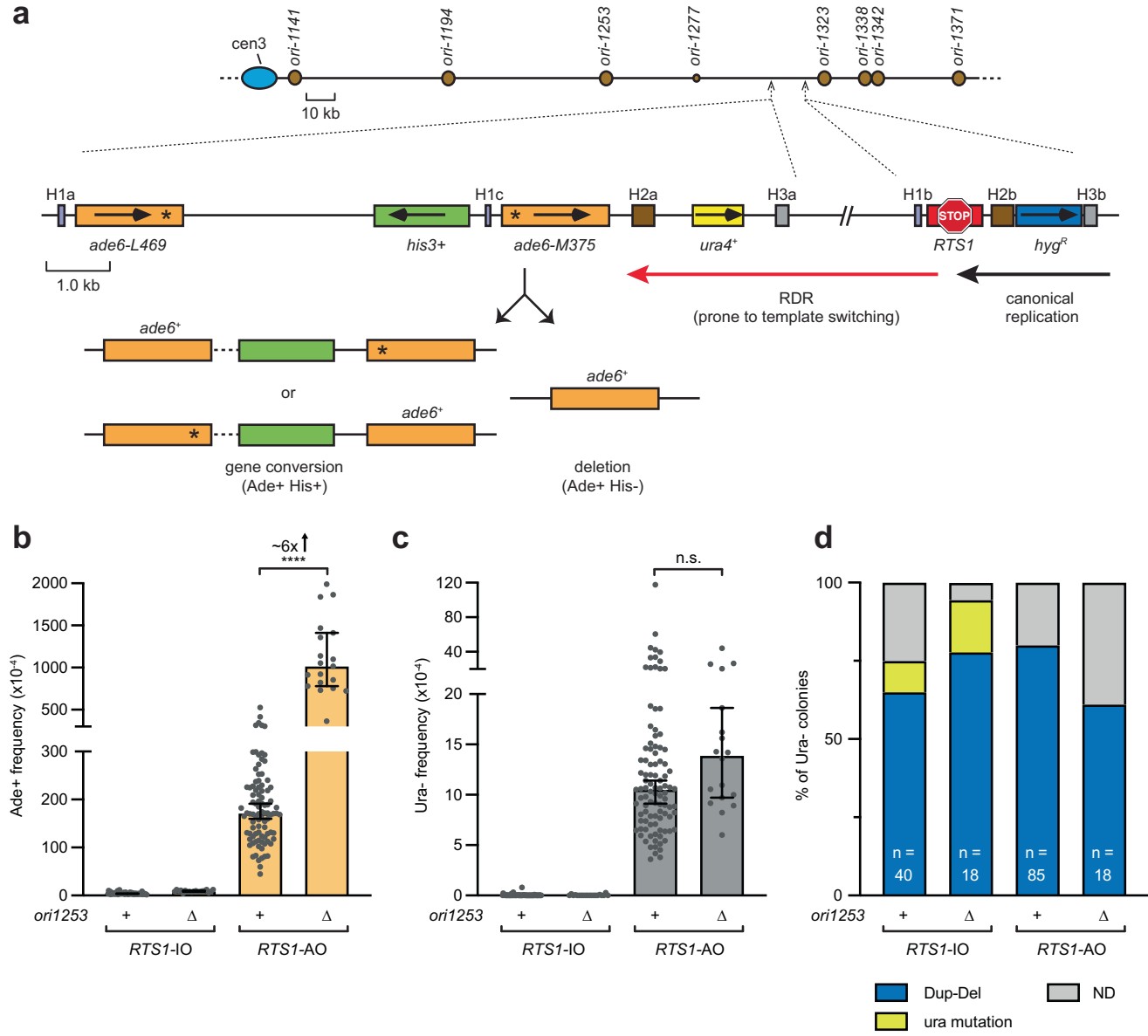

**Fig. 2 | Dup-Del formation is not constrained by oncoming DNA replication.**
**a** Genetic reporter for measuring RDR-associated template switching. The diagram shows the *ade6* direct repeat reporter adjacent to *ura4* and downstream of *RTS1* located on chromosome 3, and the two classes of Ade+ recombinants formed by template switching. The marker genes are indicated by the orange, green, yellow and blue rectangles with the arrows indicating the direction of transcription. The asterisks in *ade6-L469* and *ade6-M375* indicate the position of loss-of-function mutations. The *RTS1* RFB is indicated by the red stop symbol. The H1a/b/c, H2a/b and H3a/b sequences are indicated by the purple, brown and grey boxes, respectively. **b** Frequency of spontaneous (*RTS1*-IO) and *RTS1*-AO-induced Ade+ recombinants in the indicated strains. **c** Frequency of spontaneous (*RTS1*-IO) and *RTS1*-AO-induced Ura- colonies in the indicated strains. Data in Panels **b** and **c** are presented as median values ± 95% confidence interval with individual data points shown as grey dots. *P* values were calculated by the Kruskal-Wallis test with Dunn's multiple comparisons post-test. **** *p* value < 0.0001; n.s. not significant (*p* value > 0.9999). The data are also reported in Supplementary Data 1, which includes the strain numbers, the number of colonies tested for each strain (*n*) and *p* values. Further details of the statistical analysis are reported in Supplementary Data 2. **d** Percentage of Ura- colonies containing a Dup-Del or putative mutation in *ura4/ura5* determined by colony PCR. The data relate to the data in Panel **c**. ND indicates a failure of one or both diagnostic PCRs, and n indicates the number of independent FOA-resistant colonies tested. Source data are provided as a Source Data file.

formation, both Msh2-Msh3/Msh6 and Mlh1 seem to be equally important.

## Ku70 is a barrier to Dup-Del formation

Ku70, which is best known for its role in NHEJ, binds the free DNA end at a reversed replication fork where it acts as a barrier to DNA resection by Exo1 and aids recruitment of the ssDNA binding protein RPA and Rad51[15,16]. The estimated frequency of *RTS1*-AO-induced Dup-Dels increases by ~6.3-fold in a *ku70Δ* mutant (Fig. 6 and Supplementary Data 1). Similarly, in a *msh6Δ ku70Δ* double mutant Dup-Dels increase by ~7.5-fold compared to a *msh6Δ* single mutant (Supplementary Fig. 6 and Supplementary Data 1). These data indicate that Ku70 is a barrier to Dup-Del formation. As loss of Rad51 also causes a marked increase in Dup-Dels, Ku70's role in suppressing them may be explained by its recruitment of Rad51 to the reversed fork, which helps to protect the fork from excessive Exo1 activity[16,34]. Concordant with this idea, the increase in Dup-Dels in a *rad51Δ* mutant is partly dependent on Exo1 (Supplementary Fig. 7 and Supplementary Data 1).

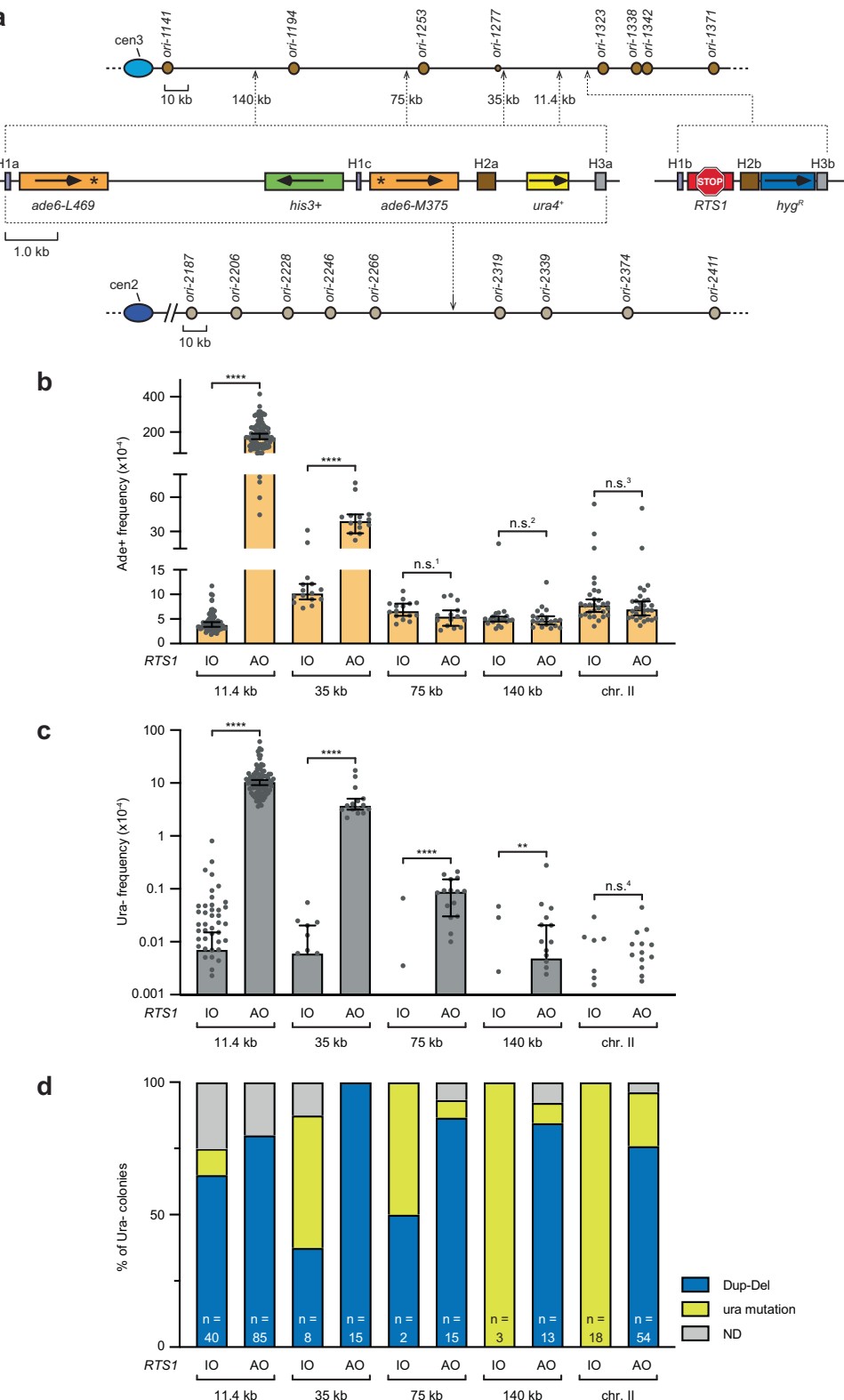

## DNA helicases with pro- and anti-Dup-Del activities

To further investigate the genetic requirements of Dup-Del formation, we screened several DNA helicase mutants that have previously been shown to affect *RTS1*-AO-induced recombination[22,38–41]. Despite causing increases in *RTS1*-AO-induced ectopic recombination and RDR-associated template switching[22,38,39], deletion of either *fbh1* or *srs2*, which encode Rad51 disruptases, or *rqh1*, which encodes a RecQ family helicase, had no significant effect on the frequency of *RTS1*-AO-

induced Dup-Dels (Fig. 5, Supplementary Fig. 8 and Supplementary Data 1). In contrast, deletion of *fml1*, which encodes an orthologue of the human tumour suppressor FANCM[42–44], or mutations in the Pif1 family DNA helicase Pfh1 that cause its nuclear depletion[45], had a marked effect on Dup-Del formation (Fig. 6 and Supplementary Data 1). With the *fml1Δ* mutant, *RTS1*-AO-induced Dup-Dels increased by ~3.4-fold, whereas with *pfh1-m21* and *pfh1-mt\** they decreased by ~7.6- and ~19-fold, respectively, which correlates with the extent of

**Fig. 3 | Dup-Del formation between distant genomic sites. a** Diagram showing the different genomic locations of the *ade6* direct repeat reporter and adjacent *ura4* gene used to investigate the effect of distance from *RTS1* on Dup-Del formation and template switching. The replication origins on chromosome 3 are indicated by brown circles, whereas those on chromosome 2 are indicated by grey circles. The marker genes are indicated by the orange, green, yellow and blue rectangles with the arrows indicating the direction of transcription. The asterisks in *ade6-L469* and *ade6-M375* indicate the position of loss-of-function mutations. The *RTS1* RFB is indicated by the red stop symbol. The H1a/b/c, H2a/b and H3a/b sequences are indicated by the purple, brown and grey boxes, respectively. **b** Frequency of spontaneous (*RTS1*-IO) and *RTS1*-AO-induced Ade+ recombinants in the indicated strains. **c** Frequency of spontaneous (*RTS1*-IO) and *RTS1*-AO-induced Ura- colonies in the indicated strains. Data in Panels **b** and **c** are presented as median values ± 95% confidence interval with individual data points shown as grey

dots. Note that some of the individual data points for the experiment in **c** are zero and, therefore, do not appear on this log-scale graph. *P* values were calculated by the Mann-Whitney test (two-tailed) or Unpaired t test (two-tailed) as indicated in Supplementary Data 1 and Supplementary Data 2. **** *p* value < 0.0001; ** *p* value 0.0036; n.s. not significant (n.s.[1] *p* value 0.1421; n.s.[2] *p* value 0.6989; n.s.[3] *p* value 0.3158; n.s.[4] *p* value 0.1137). The data are also reported in Supplementary Data 1, which includes the strain numbers, the number of colonies tested for each strain (*n*) and *p* values. Further details of the statistical analysis are reported in Supplementary Data 2. **d** Percentage of Ura- colonies containing a Dup-Del or putative mutation in *ura4/ura5* determined by colony PCR. The data relate to the data in Panel **c**. ND indicates a failure of one or both diagnostic PCRs, and n indicates the number of independent FOA-resistant colonies tested. Source data are provided as a Source Data file.

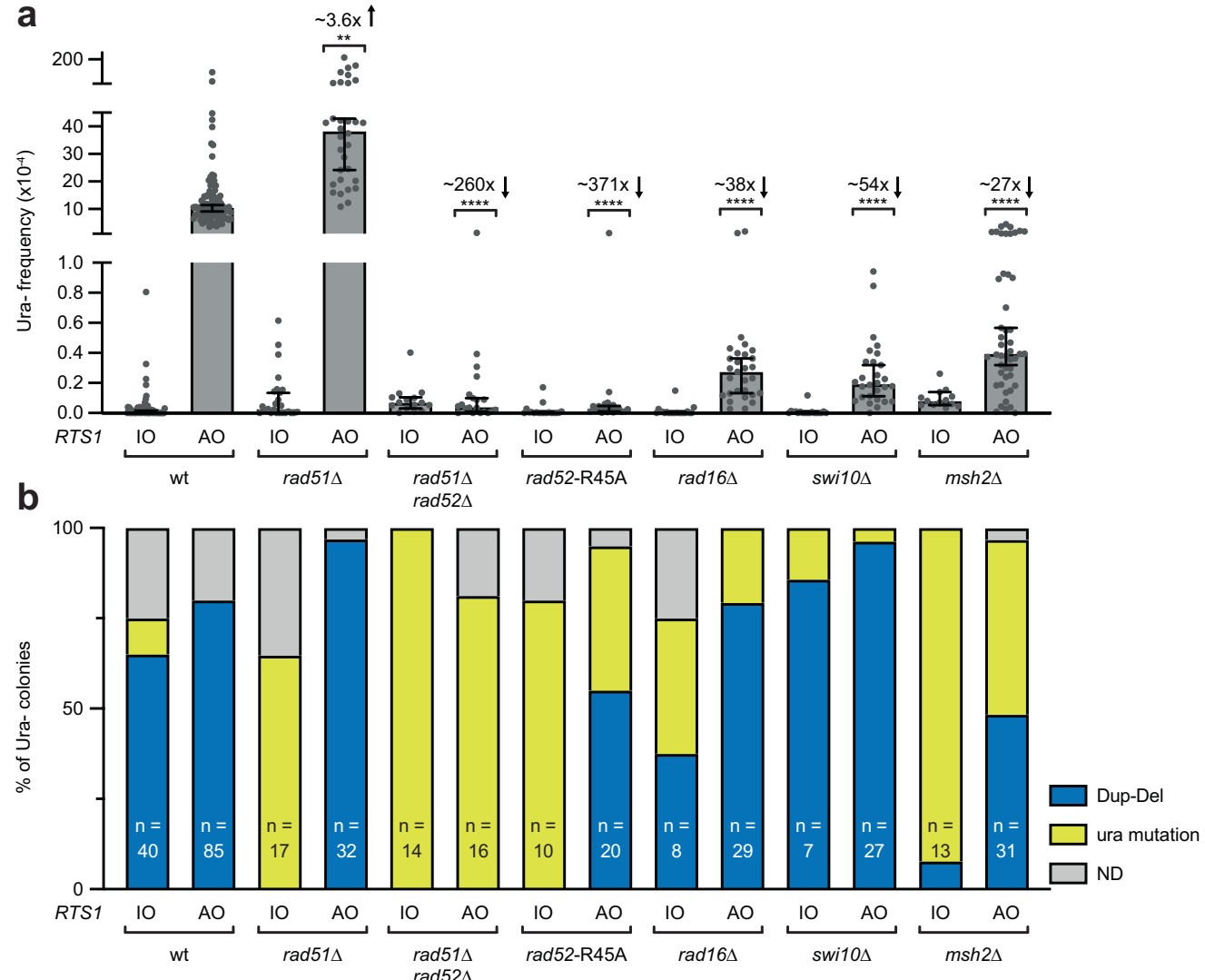

**Fig. 4 | Dup-Del formation has similar genetic requirements as TGR.**
**a** Frequency of spontaneous (*RTS1*-IO) and *RTS1*-AO-induced Ura- (FOA resistant) colonies in the indicated strains. Data are presented as median values ± 95% confidence interval with individual data points shown as grey dots. Fold changes are relative to the *RTS1*-AO wild-type strain. *P* values are for the comparison to the *RTS1*-AO wild-type strain and were calculated by the Kruskal-Wallis test with Dunn's multiple comparisons post-test. **** *p* value < 0.0001; ** *p* value 0.0028. The data

are also reported in Supplementary Data 1, which includes the strain numbers, the number of colonies tested for each strain (*n*) and *p* values. Further details of the statistical analysis are reported in Supplementary Data 2. **b** Percentage of FOA-resistant colonies containing a Dup-Del or putative mutation in *ura4/ura5* determined by PCR. The data relate to the data in Panel **a**. ND indicates a failure of one or both diagnostic PCRs, and n indicates the number of independent FOA-resistant colonies tested. Source data are provided as a Source Data file.

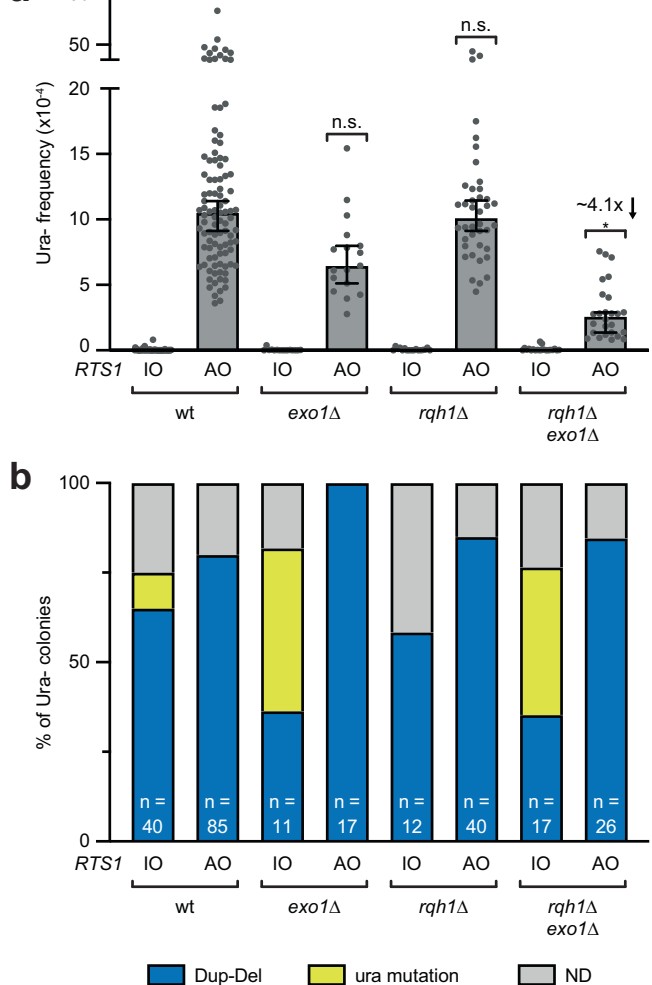

**Fig. 5 | Long-range DNA end resection is required for efficient Dup-Del formation. a** Frequency of spontaneous (*RTS1*-IO) and *RTS1*-AO-induced Ura- (FOA resistant) colonies in the indicated strains. Data are presented as median values ± 95% confidence interval with individual data points shown as grey dots. Fold change is relative to the *RTS1*-AO wild-type strain. *P* values are for the comparison to the *RTS1*-AO wild-type strain and were calculated by the Kruskal-Wallis test with Dunn's multiple comparisons post-test. * *p* value 0.0256; n.s. not significant (*p* value > 0.9999). The data are also reported in Supplementary Data 1, which includes the strain numbers, the number of colonies tested for each strain (*n*) and *p* values. Further details of the statistical analysis are reported in Supplementary Data 2. **b** Percentage of FOA-resistant colonies containing a Dup-Del or putative mutation in *ura4/ura5* determined by colony PCR. The data relate to the data in Panel **a**. ND indicates a failure of one or both diagnostic PCRs, and n indicates the number of independent FOA-resistant colonies tested. Source data are provided as a Source Data file.

nuclear depletion caused by the *m21* and *mt\** mutations[45]. Deleting *fml1* in a *msh6Δ* mutant background caused a further ~5-fold increase in estimated Dup-Dels compared to a *msh6Δ* single mutant, and we also confirmed that this heightened level of Dup-Dels was dependent on Rad16 (Supplementary Fig. 6 and Supplementary Data 1). Altogether these data indicate that Fml1 has an anti-Dup-Del activity, whereas Pfh1 has a pro-Dup-Del activity.

### RDR can generate longer tracts of over-replicated DNA for Dup-Del formation

Our finding that the duplicated DNA can include the genomic region downstream of *RTS1* suggested that RDR can create a longer tract of over-replicated DNA (Fig. 1f and Supplementary Fig. 3). However, by

Southern blot analysis we had observed only a few examples of the Dup-Del 2 and Dup-Del 3 rearrangements and, therefore, it was unclear how common they were. To get a better measure of how frequently the sequence downstream of *RTS1* (H1b) was used for Dup-Del formation, we modified our genetic assay so that the Dup-Del containing both upstream and downstream over-replicated DNA (Dup-Del 3.1) could be distinguished from Dup-Del 1.1 by the loss of an *ade6+* gene (Fig. 7a). With this modified assay, the overall frequency of *RTS1*-AO-induced Ura- colonies was similar as in the original assay and most of the tested colonies were confirmed as Dup-Dels by PCR (Fig. 7b, c, and Supplementary Table 2). The majority of these Dup-Dels (~95%) had retained the *ade6+* gene and were therefore deemed to be type 1.1 (Fig. 7b, d, and Supplementary Table 2). However, ~5% had lost the *ade6+* gene indicating that they were Dup-Del 3.1 s (Fig. 7b, d, and Supplementary Table 2). These data confirm that the H1b and H3b sequences can be used to mediate the replacement of *ade6* and *ura4* by duplicated copies of *RTS1* and *hyg^R* (Fig. 7a). The duplication of *RTS1*, together with the H3b sequence downstream of it, suggests that DRFT can occur after RDR has initiated and extended the leading strand of the collapsed replication fork beyond the barrier (Fig. 8). As discussed below, this has the potential to create longer tracts of over-replicated DNA for assimilation at nearby or distant genomic sites.

## Discussion

It has been suggested that over-replication of DNA could occur from a canonical replication fork converging with a reversed fork[8,9]. Consistent with this idea, we have shown that a RFB, where fork reversal and convergence are common, strongly induces proximal and distal gene duplications. Our data indicate that the majority of these duplications are unlikely to arise from RDR-associated template switching or Rad51-mediated multi-invasion from a reversed/unwound fork. Therefore, by a process of elimination, and the finding that Dup-Del formation has similar genetic requirements as TGR in *S. cerevisiae*, we propose that the majority of RFB-induced gene duplications detected in our assays arise from DRFT (Fig. 8). There are three versions of this model that each share the following key features: 1) replication fork reversal at a RFB and recruitment of HR proteins, which may extend the 3′-ended strand of the regressed arm by RDR; 2) merging of the reversed fork with an incoming canonical replication fork; 3) nucleolytic release of the over-replicated DNA from the site of fork convergence; 4) integration of the over-replicated DNA fragment at an ectopic site via a TGR-like mechanism. The three versions of this model differ in the timing and mode of release of the over-replicated DNA. In Version 1, the over-replicated DNA is excised from the reversed fork during fork convergence and prior to its integration at an ectopic site. In Version 2, insertion of the over-replicated DNA at the ectopic site occurs prior to its release from the site of fork convergence, and in Version 3 the reversed fork is resolved into DNA flaps during fork convergence, which are then excised and assimilated at the ectopic site.

Common to each of these variants of the DRFT model is a requirement for fork convergence at a site of fork reversal, which establishes a region of over-replicated DNA. A key finding that suggests that fork convergence is necessary for Dup-Del formation is the failure of *ori-1253* deletion to increase Dup-Del frequency. From previous work, we know that delaying the oncoming replication fork by deleting *ori-1253* results in an overall increase in recombination at *RTS1* as determined by: 1) a greater proportion of cells exhibiting a Rad52 focus at the barrier; 2) a corresponding doubling of recombination between direct repeats flanking the barrier; and 3) an increase in RDR-associated template switching downstream of the barrier[17,22]. Despite this overall increase in recombination activity, we see no increase in Dup-Del formation. This implies that Dup-Del formation does not simply depend on the recruitment of Rad52 to the *RTS1* barrier and the general promotion of recombination – it indicates that another factor is required, which is most likely fork convergence. Deleting *ori-1253* extends the

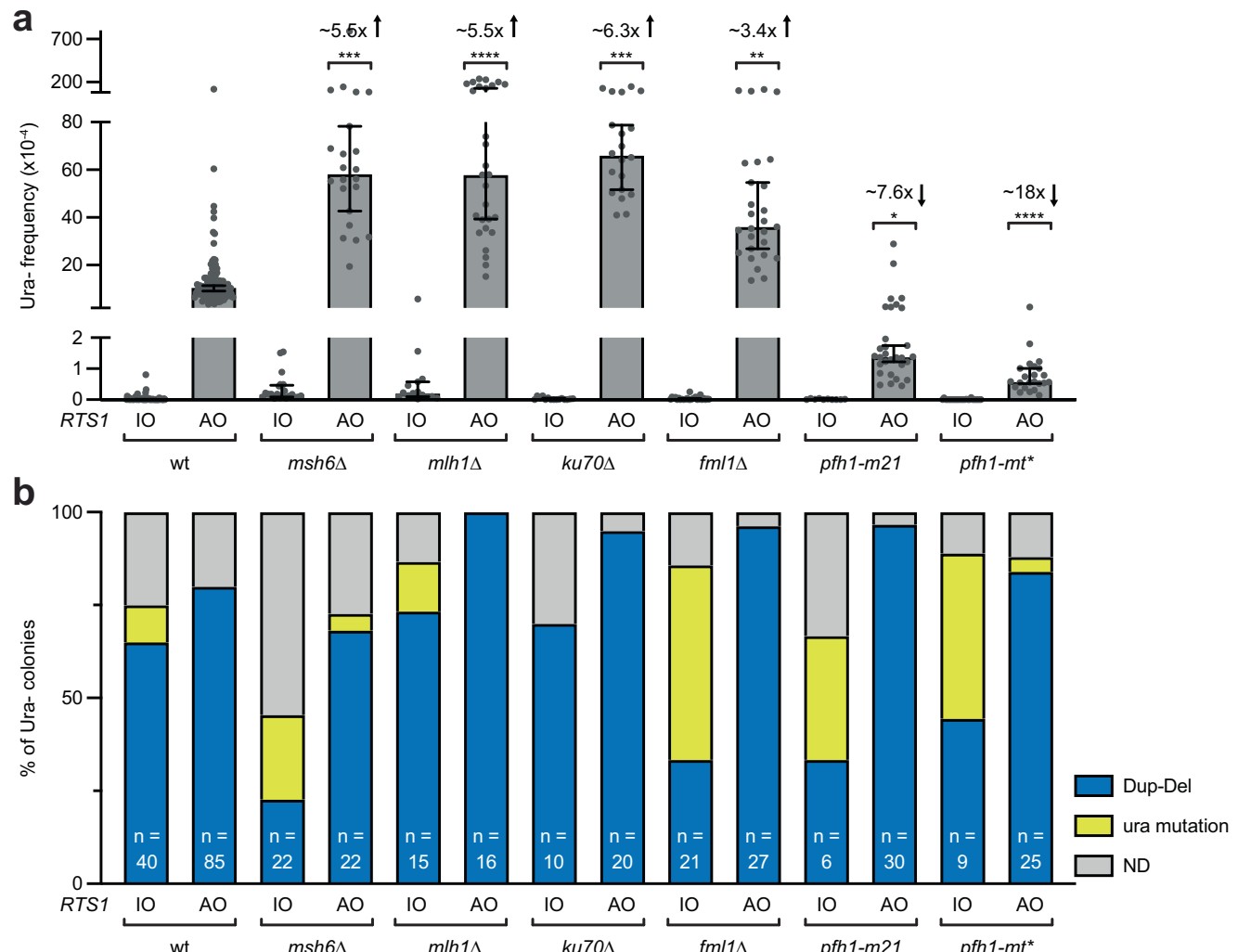

**Fig. 6 | Pro- and Anti-Dup-Del factors. a** Frequency of spontaneous (*RTS1*-IO) and *RTS1*-AO-induced Ura- (FOA resistant) colonies in the indicated strains. Data are presented as median values ± 95% confidence interval with individual data points shown as grey dots. Fold changes are relative to the *RTS1*-AO wild-type strain. *P* values are for the comparison to the *RTS1*-AO wild-type strain and were calculated by the Kruskal-Wallis test with Dunn's multiple comparisons post-test. **** *p* value < 0.0001; *** *msh6*Δ *p* value 0.0003; *** *ku70*Δ *p* value 0.0002; ** *p* value 0.003; * *p* value 0.0075. The data are also reported in Supplementary Data 1, which includes the strain numbers, the number of colonies tested for each strain (*n*) and *p* values. Further details of the statistical analysis are reported in Supplementary Data 2. **b** Percentage of FOA-resistant colonies containing a Dup-Del or putative mutation in *ura4/ura5* determined by colony PCR. The data relate to the data in Panel **a**. ND indicates a failure of one or both diagnostic PCRs, and n indicates the number of independent FOA-resistant colonies tested. Source data are provided as a Source Data file.

window of time between the first fork encounter with *RTS1* and fork convergence in the majority of cells[17]. However, the duration of this window will vary from cell to cell depending on which of the telomere proximal origins fire so, whilst in some cells there will be sufficient time for RDR to initiate and progress downstream of the barrier, in other cells it will only be sufficient to enable recruitment of Rad52. Therefore, we think that reductions in Dup-Del formation, that would result from fork convergence at sites downstream of *RTS1*, are offset by the overall increase in the number of cells in which fork reversal and Rad52 recruitment occurs prior to fork convergence. This explains why deleting *ori-1253* does not result in a reduction in Dup-Del frequency.

It has been proposed that over-replication during fork merging in eukaryotes would only be a transient feature of the termination process as any excess DNA would be rapidly degraded by nucleases such as Dna2[8,9]. Only in the absence of these nucleases would the over-replicated DNA persist and have a chance of integrating at an ectopic site[8]. Indeed, in the bacterium *Escherichia coli*, transient over-replication appears to be the norm at termination sites and several DNA nucleases and helicases are required to remove the excess DNA to

prevent it from causing pathological outcomes such as re-replication[46–51]. We have shown that, even with a full complement of nucleases and helicases, over-replication at a site-specific RFB can result in gene duplication in *S. pombe*. We suspect that the failure to efficiently degrade the extra DNA is due to it being bound and, therefore, protected by Rad52. If true, then recruitment of Rad52 to the reversed fork prior to replication termination may be a prerequisite for DRFT. The Ku70-Ku80 heterodimer has been shown to bind the reversed replication fork[15,16], and we suspect that, in addition to being a barrier to Exo1 resection, it delays the recruitment of Rad52 to provide more time for resolution of the blocked fork under conditions where fork merging will generate only transient over-replication.

Whilst we have detected inter-chromosomal Dup-Del events, Dup-Del frequency does decline with genomic distance between donor and recipient sites. Several factors could contribute to this distance effect, including 1) the half-life of the over-replicated DNA; 2) chromosomal organisation within the nucleus[52]; 3) anchorage of the collapsed replication fork at the nuclear pore complex[53]; 4) the compartmentalisation of the over-replicated DNA into a nuclear condensate[54,55]; 5) the

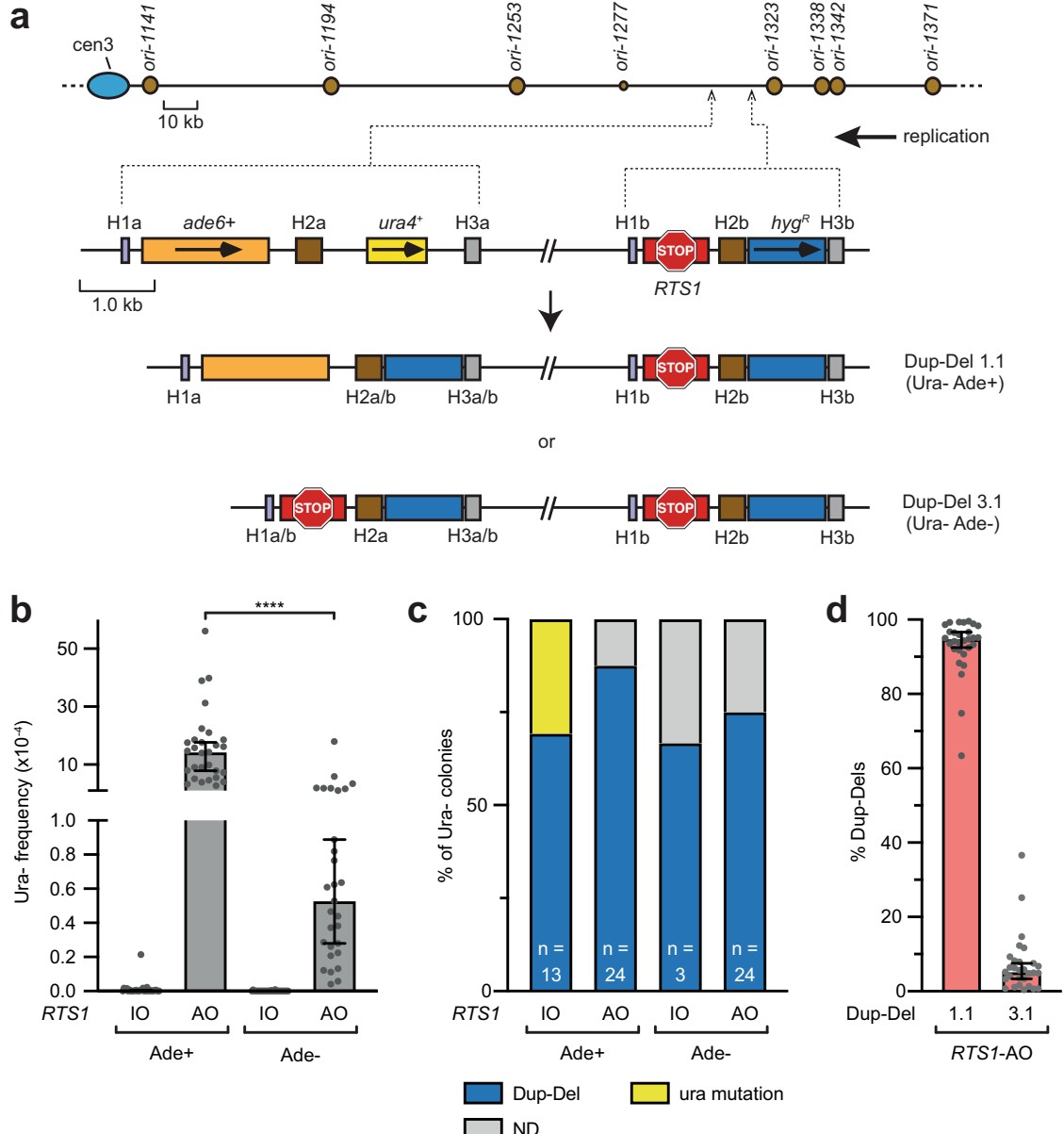

**Fig. 7 | Genomic DNA downstream of a RFB can also be over-replicated.**
**a** Diagram of the modified Dup-Del reporter and two classes of Dup-Del. The marker genes are indicated by the orange, yellow and blue rectangles with the arrows indicating the direction of transcription. The *RTS1* RFB is indicated by the red stop symbol. The H1a/b, H2a/b and H3a/b sequences are indicated by the purple, brown and grey boxes, respectively. **b** Frequency of spontaneous (*RTS1*-IO) and *RTS1*-AO-induced Ura- (FOA resistant) colonies in the indicated strains containing the modified Dup-Del reporter shown in Panel **a**. Data are presented as median values ± 95% confidence interval with individual data points shown as grey dots. *P* value = <0.0001 (****) calculated from log transformed values by the

Unpaired *t*-test (two-tailed). The data are also reported in Supplementary Table 2, which includes the strain numbers, the number of colonies tested for each strain (*n*) and *p* values. Further details of the statistical analysis are reported in Supplementary Data 2. **c** Percentage of Ura- colonies containing a Dup-Del or putative mutation in *ura4/ura5* determined by PCR. ND indicates a failure of one or both diagnostic PCRs, and n indicates the number of independent FOA resistant colonies tested. **d** Percentage of Dup-Del 1 and Dup-Del 3a amongst total *RTS1*-AO-induced Dup-Dels. Data are presented as median values ± 95% confidence interval with individual data points shown as grey dots. The values are derived from the data in Panel **b**. Source data are provided as a Source Data file.

apparent immobility of longer DNA fragments in the nucleoplasm[56,57]; 6) the possibility that the over-replicated DNA is assimilated at the ectopic site prior to its release from the site of fork convergence (Version 2 of the DRFT model). Even with the proposed protection by Rad52, we imagine that the over-replicated DNA will eventually be degraded and, therefore, has a finite time in which to encounter a genomic site where it can be integrated. Therefore, if the movement of the DNA fragment within the nucleoplasm is constrained, then nearby sites will be the ones most frequently targeted. In future studies it will be important to determine whether there is a dedicated pathway for

limiting the genomic re-insertion of the over-replicated DNA via its containment and degradation.

To identify factors that govern DRFT, we screened several helicase and nuclease mutants for their effect on Dup-Del formation. We identified the FANCM orthologue Fml1 and the Pif1 family helicase Pfh1 as being anti- and pro-Dup-Del factors, respectively. Fml1 can catalyse both fork reversal and fork restoration in vitro and, therefore, if directed appropriately, could suppress Dup-Del formation by driving fork restoration[40,58,59]. Pfh1 is required for efficient fork merging which may help to resolve the reversed fork into a structure where the over-

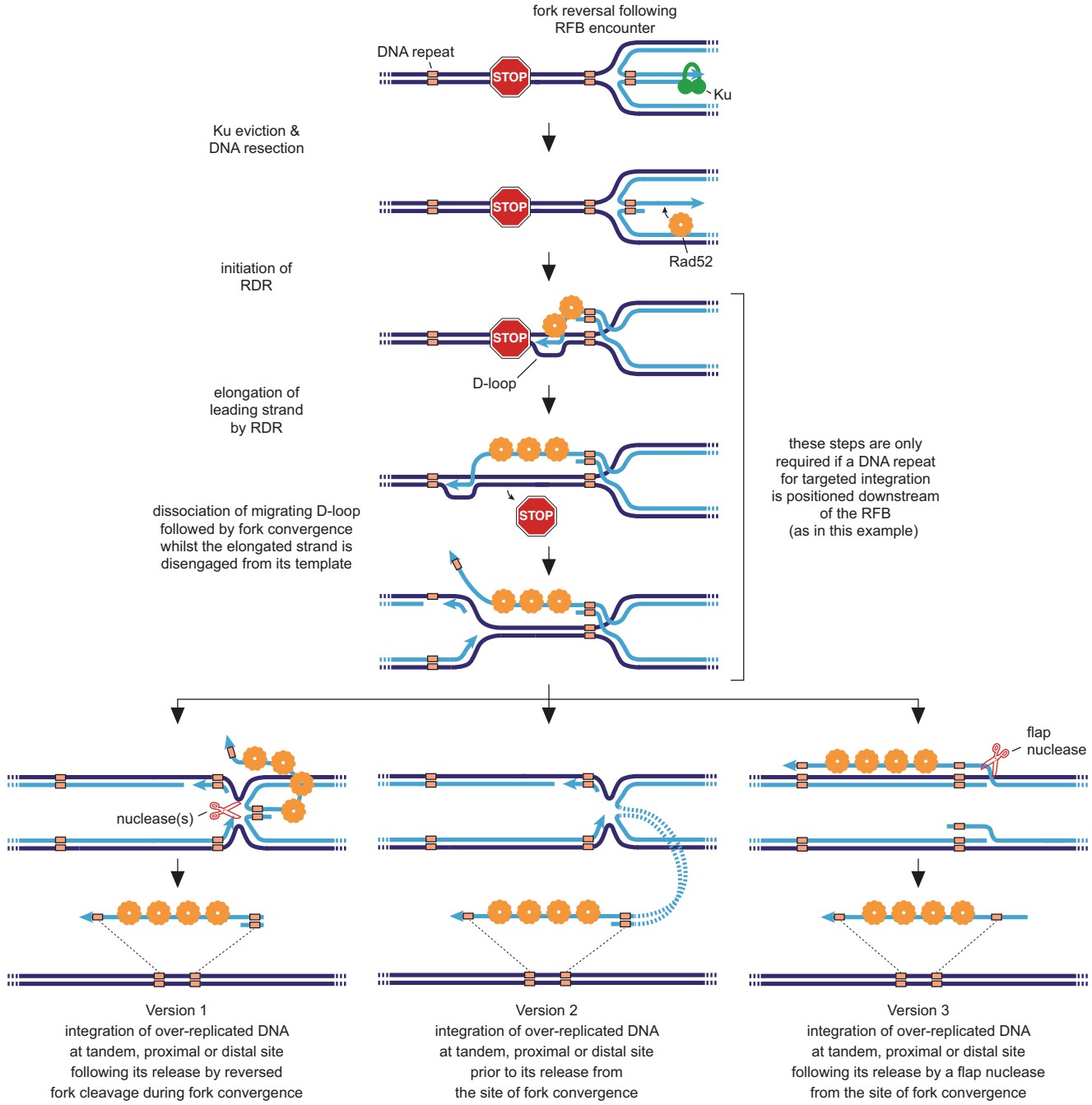

**Fig. 8 | Model for Dup-Del formation.** Three versions of the DRFT model are shown with the additional step of extending the over-replicated DNA by RDR included. As discussed in the main text, RDR is not a pre-requisite for Dup-Del formation but does offer the potential for longer tracts of over-replicated DNA to be generated. In Version 1 of the model (lefthand panel), the reversed fork is cleaved during fork convergence prior to integration at an ectopic site. In Version 2 (middle panel), the over-replicated DNA at the reversed fork is inserted at the ectopic site prior to cleavage of the reversed fork. In Version 3 (righthand panel), the over-replicated DNA is converted into 3′ and 5′ flaps during fork merging. The 3′ flap is then excised by a flap nuclease and integrated at an ectopic site by ssDNA assimilation. In each model, integration of the over-replicated DNA at an ectopic site depends on the presence of homologous DNA sequences. In humans, *Alu* elements are one example of a common repetitive DNA sequence that could mediate Dup-Del formation. Parental DNA strands are in dark blue and nascent strands are in light blue. Relevant 3′ DNA ends are indicated by the light blue arrowheads.

replicated DNA can undergo nucleolytic release[41,60,61]. We also found that a *rqh1Δ exo1Δ* double mutant was deficient in Dup-Del formation, suggesting that long-range DNA end resection promotes DRFT. In budding yeast, the deletion of the RecQ-type helicase Sgs1, together with Exo1, has been shown to either increase or decrease the efficiency of TGR depending on the assay used[62,63]. Therefore, the reduction in Dup-Del formation that we observe in a *rqh1Δ exo1Δ* double mutant could result from less efficient TGR. Alternatively, if Rqh1 and

Exo1 suppress TGR, then their role in promoting long-range DNA end resection at the stalled replication fork presumably supersedes this effect. In future studies, it will be important to distinguish between effects on the production of the over-replicated DNA and its subsequent integration into a new chromosomal site.

In each variant of the DRFT model, a structure-specific nuclease is required to liberate the over-replicated DNA from the site of fork convergence (Fig. 8). Whilst the identity of this nuclease remains

undetermined, we have excluded one potential candidate (Mus81) as Dup-Del frequency is unaltered in a *mus81Δ* mutant (Supplementary Fig. 8 and Supplementary Data 1). In our third variant of the DRFT model, the regressed arm of the reversed fork is resolved into 3' and 5' flaps during fork convergence (Fig. 8). The 3' flap, which may remain bound to Rad52, could then be excised and integrated at an ectopic site by single-strand assimilation and heteroduplex correction rather than a mechanism involving strand invasions from both ends of the over-replicated DNA[32,64,65]. In budding yeast, Rad1 and Msh2-Msh3 act to prevent single-strand assimilation[32]. If the same is true for their orthologues in fission yeast, then their role in promoting DRFT would most likely be in the removal of 3' heterologous flaps during single-strand assimilation. However, it is also possible that Rad16-Swi10 is the 3' flap nuclease that liberates the over-replicated DNA in this version of DRFT (Fig. 8)[66].

How far a fork reverses could limit the amount of DNA that is over-replicated during DRFT. In human cells, when fork reversal was induced by modest levels of genotoxic stress and visualised by electron microscopy, the average size of the regressed arm was only ~300 bp, although arms as long as ~4 kb were detected[11]. In our study, the duplicated *hyg^R* gene with flanking H2b and H3b sequences is ~1.6 kb suggesting that fork reversal at *RTS1*-AO can extend at least over this distance. If the size of the over-replicated DNA is determined solely by the extent of fork reversal, then the ability of DRFT to generate duplications larger than a few kb would seemingly be curtailed. However, our finding that DNA downstream of the RFB can be included in the region that is duplicated suggests that the over-replicated DNA can be extended by RDR (Fig. 8). As RDR is susceptible to template switching, there will be times when the elongating DNA strand is disengaged from its template. Fork convergence during these moments of elongating strand disengagement would result in a longer section of over-replicated DNA (Fig. 8). Moreover, as Rad52 is the main driver of template switching, it is likely to be bound to the over-replicated DNA when fork convergence occurs[19]. Our proposal, that RDR can extend the regressed arm of a reversed fork, opens up the possibility that DRFT could generate much larger and more complex DNA duplications, especially if the extended DNA strand has undergone one or more template switch events. As such, DRFT could be responsible for generating a significant subset of the structural variants that are characteristic of cancer genomes.

## Methods

### Yeast strains, media and growth conditions

*S. pombe* strains are listed in Supplementary Data 3 and are available from the corresponding author upon reasonable request. Standard protocols were used for the growth and genetic manipulation of *S. pombe*[67]. Derivatives of recombination reporter strains carrying the indicated gene/replication origin deletion(s) were obtained from genetic crosses. The strains with reporters at different genomic locations (Fig. 3) were obtained by a one-step marker swap protocol in which *kanMX6* within a previously constructed lab strain was replaced by *ura4MX4*[68]. Yeast strains were grown on/in complete and minimal media at 30 °C as required. The complete and minimal media were yeast extract with supplements (YES) and Edinburgh minimal medium plus 3.7 g/l sodium glutamate (EMMG) and appropriate amino acids (225 mg/l), respectively. To maintain the integrity of the various genetic reporters, strains carrying them were grown on EMMG lacking histidine and/or uracil as appropriate. Additionally, strains carrying *ade6-L469 – ade6-M375* genetic reporters were grown on media supplemented with low levels of adenine (10 mg/l) to distinguish non-recombinant colonies (red) from Ade+ recombinants (white). Ade+ recombinants were selected on YES lacking adenine and supplemented with 200 mg/l of guanine to prevent uptake of residual adenine. Ura- recombinants were selected on YES containing 1.5 g/l of 5-FOA.

### Recombination assays

To determine the frequency of FOA resistant and Ade+ colonies, strains containing relevant reporter constructs were grown for 4 – 5 days on YES plates at 30 °C. Similar-sized "initial" colonies were then suspended in 0.35 ml of sterile water and serially diluted. Appropriate dilutions were plated on YES containing low adenine (10 mg/l) (YES/LA), YES plus 5-FOA (YES + FOA), and YES minus adenine plus guanine (YES-ade+gua) plates, except for the experiment in Fig. 7, where cells were plated on YES/LA and YES/LA plus 5-FOA (YES/LA + FOA) plates. Colonies on YES/LA, YES + FOA and YES/LA + FOA plates were counted after 4 days incubation at 30 °C, whilst those on YES-ade+gua plates were counted after 6 days. Colonies were counted using a Protos 3 automated colony counter with Protos 3 Version 1.2.4.0 software (Synoptics Ltd), and the percentage of FOA-resistant colonies and Ade+ recombinants amongst total viable (colony forming) cells was determined by comparing the number of colonies on the YES/LA plate with those on the YES + FOA or YES/LA + FOA and YES-ade+gua plates, respectively. The percentage of deletions and gene conversions amongst Ade+ recombinants was determined by replica plating the YES-ade+gua plates onto EMMG plates lacking histidine and adenine. The frequency of Ade+ recombinants amongst FOA-resistant colonies was determined by replica plating FOA plates onto YES/LA and counting the number of red (Ade-) and white (Ade+) colonies. For the data in Fig. 7, Dup-Del 1.1 and Dup-Del 3.1 recombinants were distinguished by the colour of the FOA resistant colony on YES/LA + FOA plates (Dup-Del 1.1 = white; Dup-Del 3.1 = red). Each strain was assayed at least twice with between 5 to 20 "initial colonies" analysed in each experiment. Recombination frequencies were determined using the method of the median to prevent skewing of the data by "jackpot" events, where a single recombination event at an early stage in the growth of the colony can give rise to many recombinant cells. To determine the percentage of Dup-Del recombinants amongst FOA resistant colonies, PCR analysis was conducted on randomly selected FOA resistant colonies. To avoid analysing colonies from clonal populations, no more than one FOA resistant colony from each "initial colony" was analysed, except for strains MCW9350 and MCW9351 where all FOA resistant colonies were analysed. The PCR analysis utilised primer sets A (oMW1620: AATACTAGTGCGCTG-TAACTTACCTAC and oMW1985: CACATCCGAACATAAACAAC) and B (oMW1628: TTAATAACTAGTCTTAATATTGC and oMW1985: CACAT CCGAACATAAACAAC) (Fig. 1a). A minority of the colony PCRs failed to yield a DNA band and, therefore, the Dup-Del status of these colonies remained undetermined (indicated by ND in the relevant Figures). However, in approximately 77% of all ND cases only the PCR for primer set B failed and, therefore, we could determine whether the *ura4* gene had been retained or replaced by *hyg^R*. Approximately 83% of these colonies were the latter type where the *ura4* gene had been replaced by *hyg^R*. This information is recorded in the Source Data file, which includes the raw data for the recombination assays and all PCR analysis.

### Statistical analysis

Analysis of recombination data was performed using Excel for Mac Version 16.74 (Microsoft®) and GraphPad Prism Version 9.3.1 (Graph-Pad Software, San Diego, CA). Recombination frequencies were analysed for normal distribution using the Shapiro-Wilk test. Not all data passed this test and, therefore, most recombination frequencies were compared using a Kruskal-Wallis test (one-way ANOVA on ranks) with a Dunn's multiple comparisons post-test or a two-tailed Mann Whitney test. These are non-parametric statistical tests and, therefore, do not require the data to be normally distributed. Where appropriate, recombination values were log-transformed and analysed using the Unpaired t-test. P-values are reported in the figures, Supplementary Data 1, and Supplementary Table 2. Details of the statistical analysis of the data in Figs. 1–7, Supplementary Figures 4, 6, 7 and 8, Supplementary Data 1, and Supplementary Table 2 are summarized in Supplementary Data 2.

## Colony PCR

Single colonies from FOA plates were resuspended in 30 μl of sterile Milli-Q water and frozen for later analysis. 0.5 μl of the thawed cell suspension was added to a PCR mix consisting of 3 μl 10x ThermoPol buffer (New England Biolabs, M0267X), 0.6 μl 10 mM dNTPs (ThermoFisher Scientific, R0182); 0.6 μl 10 μM oMW1985, 0.6 μl 10 μM oMW1620/1628, and 0.15 μl of *Taq* DNA Polymerase (New England Biolabs, M0267X) in a final total volume of 30 μl. The mixtures were assembled in PCR tubes on ice and then transferred to a PCR machine (C1000 Touch Thermal Cycler, BioRad) with the block pre-heated to 98 °C. The reactions were then heated at 98 °C for 5 min before cycling through 38 cycles of 95 °C for 30 s, 50 °C for 30 s, and 68 °C for 2 min. Upon completion of the final cycle, reactions were heated for a further 5 min at 68 °C before being placed on ice. The products of the reactions were run on 1% agarose gels, which were stained with ethidium bromide and analysed using a ChemiDoc XRS+ Imaging System (BioRad) and Image Lab software (BioRad, Version 6.1.0 build 7).

## Preparation of genomic DNA for Dup-Del analysis

Single colonies from YES + FOA plates were patched onto fresh YES + FOA plates and grown at 30 °C. The patched yeast cells were then used to inoculate 100 ml YES broth in shake flasks and grown to saturation at 30 °C. Cells were harvested by centrifugation and resuspended in 5 ml of CPES buffer (40 mM citric acid, 120 mM Na$_2$HPO4, 400 mM EDTA, 1.2 M sorbitol) plus 15 mg of Zymolyase 20 T (MP Biomedicals, 08320922). The mixture was incubated at 37 °C for 30–60 min until cells were spheroplasted. The spheroplasted cells were then collected by centrifugation and resuspended in 15 ml of 5x TE (50 mM Tris-HCl pH 7.5, 5 mM EDTA). 1.5 ml of a 10% SDS solution was then added to the mixture to lyse the cells. Following cell lysis, 5 ml of 5 M potassium acetate was added and the mixture was chilled on ice for 30 min before centrifuging at 3200x *g* for 15 min. The supernatant was then carefully transferred to a fresh tube and the DNA precipitated by addition of 20 ml of ice-cold isopropanol. Following incubation for 5 min at −20 °C, the sample was centrifuged at 3200x *g* for 30 min. The resulting pellet was dried and resuspended in 3 ml of 5x TE to which RNase A (Qiagen, 1007885) was added to a final concentration of 20 μg/ml. The mixture was then incubated for 2 h at 37 °C before mixing in an equal volume of phenol:chloroform:isoamyl alcohol (25:24:1 v/v; ThermoFisher Scientific, 15593031). Following centrifugation, the upper aqueous phase was transferred to a fresh tube and the DNA was re-precipitated by addition of 0.3 ml of 3 M sodium acetate plus 3 ml of isopropanol, and incubation overnight at −20 °C. The precipitated DNA was collected by centrifugation and washed three times with 5 ml of ice cold 70% ethanol before being dried and resuspended in 0.2 ml of 1x TE.

## 2DGE of replication intermediates

Yeast strains (MCW9374 and MCW9235) were grown in 500 ml of EMMG supplemented with leucine, adenine and arginine in baffled shake flasks at 30 °C until they attained a cell density of ~1 ×10$^7$ cells/ml. At this point, the cells were treated with 0.1% sodium azide and harvested by centrifugation. Cell pellets were washed in 20 ml 50 mM EDTA and stored in two equal sized aliquots at −80 °C to await further processing. Cells from one aliquot were defrosted on ice and gently resuspended in an equal volume of CPES buffer to which *Trichoderma harzianum* lysing enzymes (10 μl/ml of a 250 mg/ml solution; Sigma-Aldrich, L1412), *Arthrobacter luteus* lyticase (20 μl/ml of a 25 mg/ml solution; Sigma-Aldrich, L4025), Zymolyase 20 T (20 μl/ml of a 100 mg/ml solution; MP Biomedicals, 08320922), and DTT (10 μl/ml of a 1 M solution; Sigma-Aldrich, D9779) were added. The suspension was then incubated at 37 °C for up to 2 h until spheroplasting of the cells was achieved. The spheroplasted cells were then mixed with an equal volume of molten low-melting point agarose (2%; Thermo Scientific, R0801) in 0.25 M EDTA and 1.2 M sorbitol. The cell/agarose mixture was pipetted into ~30 plug molds (BioRad, 1703713) and allowed to set at 4 °C for 15 min. The plugs were

then removed from the molds and incubated for 36 h at 50 °C in 3 ml of NDS-PK buffer (10 mM Tris-HCl pH 7.5, 1% N-lauroylsarcosine, 495 mM EDTA, 0.5 mg/ml Proteinase K), with the buffer being replaced with a fresh 3 ml aliquot after 18 h. Following removal of the NDS-PK buffer, the plugs were washed for 1 h in 10 ml of 1x TE buffer (10 mM Tris-HCl pH 8.0, 1 mM EDTA) containing 0.5 mM PMSF, followed by three sequential 1-hour wash steps in 20 ml of 1x TE. The plugs were then kept overnight at 4 °C in 1x TE. For all the following steps, DNA LoBind tubes (Eppendorf: 1130122232; 0030122208; 022431021) were used. The plugs were washed in 20 ml of Milli-Q water for 30 min and then 10 ml of 1x rCutSmart buffer (New England Biolabs, B6004S) for 1 h followed by a further hour in 5 ml of 1x rCutSmart buffer. The buffer was then removed and the plugs were melted by placing the tube at 65 °C in a water bath for 15–20 min. The tube containing the plugs was then cooled to 42 °C before adding 5 μl of Beta-Agarase I (New England Biolabs, M0392) and incubating at 42 °C for 1 h. 200 units of MfeI-HF (New England Biolabs, R3589L) plus 10 μl of RNase A (100 mg/ml; Qiagen; 1007885) were then added and the mixture was incubated overnight at 37 °C. An extra 50 units of MfeI-HF was added the next day and the incubation continued for a further 6 h. The mixture was then centrifuged at 3200x *g* for 10 min and the supernatant transferred to a fresh tube. The supernatant was then mixed with an equal volume of phenol:chloroform:isoamyl alcohol (25:24:1 v/v; ThermoFisher Scientific, 15593031), and centrifuged for 10 min. DNA in the aqueous phase was then precipitated by addition of an equal volume of isopropanol plus 0.1 volume of 3 M sodium acetate and incubation overnight at −20 °C. Following centrifugation, the DNA pellet was washed three times in 70% ethanol, dried and dissolved in 40 μl of 1x TE buffer. The sample was then mixed with 4 μl of 10 x gel loading dye (50% glycerol, 1% SDS, 0.1 M EDTA, 0.25% bromophenol blue) and run on a 0.4% agarose gel in 1x TBE buffer for 48 h at 25 V. The gel was then stained with Ethidium Bromide for 1 h and washed in water for a further hour. The lane containing the DNA was then carefully excised from the gel and incorporated into a 1.2% agarose gel in 1x TBE containing Ethidium Bromide such that the gel slice was perpendicular to the direction of electrophoresis. The gel was run at 4 °C for ~18 h at 120 V with buffer re-circularisation.

## Southern blotting

1D (Fig. 1f and Supplementary Fig. 3) and 2D (Fig. 1b) gels were treated with 0.25 M HCl to depurinate DNA, and then DNA was transferred to GeneScreen Plus membrane (Perkin Elmer, NEF988001PK) by capillary action under alkaline conditions. Following transfer, the membrane was probed with $^{32}$P-radiolabelled *hyg$^R$* probe (Rediprime II Random Prime Labelling System, Cytiva, RPN1633; and alpha-$^{32}$P dCTP, Perkin-Elmer, NEG513H250UC) in ULTRA-hyb buffer (Invitrogen, AM8669) at 42 °C. The membrane was then washed following the manufacturer's recommended procedure and exposed to a phosphor screen for up to 5 days. The phosphor screen was scanned using a FLA-3000 Phosphorimager (Fujifilm) controlled by Image Reader software (Fujifilm, Version 2.02), and the data analysed using Image Gauge software (Fujifilm, Version 4.21).

## Reporting summary

Further information on research design is available in the Nature Portfolio Reporting Summary linked to this article.

## Data availability

All data generated or analysed during this study are included in this published article and its Supplementary Information file. The recombination data generated in this study have been deposited in figshare [https://doi.org/10.6084/m9.figshare.24231703]. The replication origins indicated in Figs. 1a, 2a, 3a, 7a, and Supplementary Figures 3a and 5a are listed in OriDB (http://pombe.oridb.org). Source data are provided with this paper and can be found in figshare [https://doi.org/10.6084/m9.figshare.24231703]. Source data are provided with this paper.

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

## Acknowledgements

We thank Manisha Jalan and Steven Pilley for making pre-cursors of some strains used in this study. ChatGPT (chat.openai.com) was used to check grammar and make a few minor grammatical improvements to the text. This work was supported by grants from the Medical Research Council (MR/V009214/1 awarded to M.C.W.), and Biotechnology and Biological Sciences Research Council (BB/P019706/1 and BB/V00073X/1 awarded to M.C.W.). The funders had no role in study design, data collection and analysis, the decision to publish, or preparation of the manuscript.

## Author contributions

J.O. and C.A.M performed the experiments; J.O, C.A.M. and M.C.W. contributed to experimental design, data analysis and presentation. M.C.W. secured funding and wrote the manuscript. J.O, C.A.M. and M.C.W. edited the manuscript.

## Competing interests

The authors declare no competing interests.
