## [Peer Review File · Nature Communications]

Gene duplication and deletion caused by over-replication at a fork barrierREVIEWER COMMENTS

Reviewer #1 (Remarks to the Author):

The authors use a genetic reporter assay to study the effect of a site-specific replication fork block on the frequency of both RDR (recombination-dependent replication) events and duplication-deletion (Dup-Del) rearrangements in fission yeast. They show that a site-specific RTS1 barrier can induce high frequency Dup-Del rearrangements. The Dup-Del events described use short sequence homologies flanking the HPH drug resistance cassette inserted next to the RTS1 site to replace a *ura4* cassette flanked by the same homologies on the unreplicated side of the fork barrier. The authors claim that these events occur through a model they call DRFT (Duplication by Reversed Fork Termination), a mechanism they state is similar to targeted gene replacement (TGR) based on the genetic requirements, and are independent from RDR or Rad51-dependent multi-strand invasion. In this model, they propose that fork stalling can lead to the over-replication of DNA at a reversed fork upon convergence with an oncoming replication fork, as well as the eventual release of a DNA fragment that can then be integrated at another site in the genome using the short flanking homologies.

The data reported are intriguing but I am not completely convinced that the Dup-Del events occur by the mechanism proposed for the following reasons:

1. The data presented in Figure 2 is not consistent with the need for a converging fork. Surely if the arrival of the converging fork were to be delayed (*ori1253* deletion), it would reduce the frequency of Dup-Del events? Could another RTS1 barrier be inserted to interfere with replication initiated at centromere proximal sites to reduce fork convergence to directly test this aspect of the model?
2. The frequency of Dup-Del events is dramatically reduced as the *ura4* cassette is placed further from the RTS1 barrier and undetectable when it is on another chromosome. This distance effect suggests a cis interaction and is not consistent with release of the fragment from the stalled fork. Why does the fragment need to be released? It seems just as plausible for the resected reversed fork to invade the *ura4* cassette and cleavage to be associated with strand assimilation. The authors cite work from the Ira lab showing increased insertions at a targeted DSB in the budding yeast *dna2* mutant as support for their model. However, in that study the insertions were in trans not in cis. Moreover, the authors did not test the *dna2* conditional mutant to see if the frequency of Dup-Del events increases.
3. In support of their model, loss of Rad52 strand annealing activity, Rad16 (Sc Rad1), and Msh2 all decrease the frequency of Dup-Del formation, and all of these genes are also required for TGR in budding yeast. To my knowledge these mutants have not been shown to be defective for TGR in fission yeast. The increase in Dup-Del events in the absence of Rad51 is unexpected since TGR is decreased in HR-defective mutants. It would be of interest to determine whether TGR is Rad51-independent in fission yeast and has the same dependency on Rad16. This could be easily tested by amplifying the HPH cassette with flanking homologies and transforming the fragment (as ssDNA or dsDNA) into cells to replace the *ura4* cassette (preferably in a strain with no HPH cassette).
4. If resection is important, would an *exo1Δ rqh1Δ* be more definitive compared to using either of the single mutants?
5. Finally, the DRFT model predicts fork reversal and cleavage of the regressed arm. However, of the helicase mutants tested, only Pfh1 reduced the frequency of Dup-Del events, and there is the complication of the role of Pfh1 in termination. Are there other translocases that function in fork reversal in fission yeast that could be tested? Although Dup-Del events are reduced in the *rad16* mutant, there is no evidence that Rad16 cleaves a reversed fork and it seems more likely that it is acting at the heterology boundary during integration of the fragment. Mus81 has been shown to act at stalled and reversed forks and should also be tested in this assay.

Other comments:

6. Plotting Ura- frequency: In Figure 3C, the data are fitted to a log scale. This seems to be a much better visual representation of the data (although I'm not sure why a few of the data sets in 3C don't

have bars) than using a linear scale. In 1C, 2C, 4A, 5A, and 6B, the reader can't appreciate the spread of the data since many of the data sets hug the x-axis. This would be an easy fix and improve the figures readability significantly.

7. Inclusion of IO and AO data: While I understand the rationale for inclusion of both IO and AO data in the earlier figures to validate that the reporter works and the RTS1 block is directional, I don't see the need to include these data in Figures 4 and 5. Given that these figures are probing the genetic requirements of dup-del formation in response to replication fork stalling, it seems that the IO data doesn't contribute to the model. These data could probably be moved to the supplement. In that case, Figures 4 and 5 could be combined into one four panel figure. Alternatively/additionally, this would free up space for discussion of other mutants such as *exo1Δ* to be presented in the main text or to use Figure 6 only to describe different models for Dup-Del generation.

8. Presenting breakdown for % of Ura⁻ colonies: The n values for colonies scored for Dup-Dels or ura mutations vary significantly. While I don't think all strains tested need to have dozens of colonies scored, certainly an n value of 2 is not representative of all the events detected in these assays. There should be some minimum threshold for how many colonies are scored.

9. Figure 1A, 2A, 3A, 6A: The reporter used in these experiments as currently presented is a bit overwhelming. For example, I'm unsure as to why so many of the origins are labeled. It would be a lot simpler to only label the origins suspected to be firing through the reporter. The figure could also be simplified by removing the arrows indicating direction of transcription of the various parts of the reported since transcription is not mentioned in the text. Finally, it might be nice to have RTS1 as a stop sign or triangle to indicate that it's not simply part of the reporter; it is initiating the fork stall (as is done in the model).

10. The recombination events in this assay seem to be extremely prevalent (1 in 1,000 cells). I find this somewhat surprising given the mechanism proposed. Can the authors comment on this?

Reviewer #2 (Remarks to the Author):

The new manuscript by Oehler et al. presents a new mechanism leading to deletion/duplication. They use a fission yeast system to study genome instability. The experimental model has reporter markers URA4 and *hygR* and replication fork barrier sequence (RTS1) in between the two markers. RTS is positioned in such a way that replication forks encountering RTS can produce reversed forks within the *hygR* sequence. *hygR* sequence is flanked on both ends by 250-400 bp sequences homologous to URA4 flanking sequences and other upstream sequences and therefore can lead to recombination generating ura minus colonies that can be scored on FOA plates. Fork blocking induced such dup-del events by nearly 2000 times. When the URA4 marker was further away or even on different chromosomes dup-del recombination was also induced by fork blocking, however at lower frequencies. Recombination was also not reduced but rather increased in the absence of Rad51 excluding regular template switching mechanism. Genetic analysis shows the requirement for Rad52 annealing activity, Rad1-Rad10 nuclease, and Msh2. This is an identical gene set of genes needed for gene targeting suggesting a model where duplicated region at the reversed fork may be released and integrated elsewhere in the genome.

The strengths of this interesting work are:

Demonstration that replication fork barrier/reverse replication forks can lead to duplication-deletion and comprehensive genetic analysis of these events suggesting possible mechanism.

The manuscript is well written.

Comments

1. Could sister chromatid recombination with no crossover contribute to URA4 X *hygR* recombination? Fasullo's lab demonstrated that spontaneous inter-sister recombination could occur in *rad51* minus

mutant in budding yeast.

2. Could strand assimilation occur when over-replicated region is not yet liberated? This could explain the higher rate of dup-del with a reporter gene in proximity to RTS. Alternatively could local supercoiling change with a reversed fork that could open duplex DNA and facilitate annealing?

3. It is not tested/discussed which nuclease could be involved in the release of a duplicated region. In discussion, Rad1/Rad10 is suggested.

4. Please provide exact homology between all H sequences flanking markers and others. How many mismatches are present?

Reviewer #3 (Remarks to the Author):

The submitted study by Oehler, Morrow and Whitby utilize the well-studied RFB system, and a marker based genetic assay in *S. pombe* to investigate mechanisms that drive gene duplication events. These duplication events occur during DNA replication fork stalling and fork reversal, which the authors term duplication by reversed fork termination (DRFT) wherein a canonical replication fork merges with a stalled reversed fork resulting in the over-replication of DNA. The excess DNA, if not degraded, can then be integrated at an ectopic site via a mechanism like targeted gene replacement (TGR) and result in duplication-deletion events (dup-dels). The authors identify that the formation of these dup-dels requires Rad52, Rad16-Swi10, and Msh2. They show loss of Ku70 and Rad51 increase dup-del formation and hypothesize that their absence could result in impaired fork protection. Additionally, the helicases Fml1 and Phf1 have opposing roles in dup-del formation due to their roles in fork restoration and fork merging, respectively. Altogether, the authors suggest that DRFT could contribute to genomic instability and structural variations in the genome, including those associated with cancer genomes. While I believe this study's conclusions are supported by compelling data and presents an interesting mechanism for genomic instability and structural variation, there is room for improvement in the text and scientifically. The manuscript would be much improved with more details throughout. Of note, much of this manuscript is lacking adequate context to set up the rationale for experiments (Figure 2 and Figure 6 in particular). It would also help if the findings throughout were concluded in ways that linked back to the initiating rationale. Additionally, there are significant inconsistencies among presented data which need to be clarified before it would be suitable for publication.

Major Concerns:

1. In the manuscript, you don't discuss the possible mechanisms by which you get no amplification by PCR (ND). Have you done Southern blotting to look at the restriction fragments at your loci? Discussion and determination of this population is essential because it is not a small population (many have 20% or more ND).

a. Along these lines, you mention in line 129-130, of the 40 *ura-* colonies you tested that are in the AO condition all of them are Dup-Dels but your graph in 1d suggests that 20%-30% should be colonies where no detectable PCR occurs. Please explain the discrepancy between your data in 1d and figure 1e, f, Sup Figure 2 and 3.

2. The data in figure 1d presented in the paper a few times but is not consistent—Figure 2d has the same test population (RTS-AO + origin) and it has a completely different distribution of outcomes. The data seems to match in figure 1d and 4b. This is important because claims related to figure 2d state there is no difference between the presence or absence of the origin of replication, but if the data in 2d is to be believed then loss of the origin increases ND and eliminates mutation or URA3. Please clarify your graphs and the conclusions you can draw from these data.

3. In figure 2b and 2c you are measuring two separate phenomena: RDR-associated template switching and dup-dels. The incidence of ADE+ reversion is very high (10^{-1} or 10^{-2}). What is the co-incidence of *ura-* *ade+* cells, which would be colonies where dup-dels and template switching resulted

in mutational repair? Are Ura⁻ colonies more likely to be Ade⁺?

4. Previous work from the Carr lab (10.1038/s41467-021-21198-0) showed that RDR is completed by a δ/δ fork on the leading and lagging strand and that less processive synthesis leads to increased gaps (which could be a mechanism for microsatellite mediated instability: (10.1371/journal.pgen.1009863). Linking point 3 and 4, could remaining post-replicative gaps be sites of annealing between the cleaved reversed fork and the region of homology that will be deleted?

a. Ln 142-153 does a poor job describing the combinatorial assay (ADE⁺ reversion and ura⁻), the previous work and what your data output means in terms of your developing model. Please add additional context to the description of the assay and consider adding a replication fork model demonstrating the possible outcomes.

5. The increase in dup-del rate in the rad51 Δ mutant is surprising and your hypothesis about Ku and Rad51 is intriguing (Ln 215-216). What happens in the rad51 Δ ku Δ double mutant in terms of dup-dels?

6. It's difficult to ascertain whether movement to 35 kb changes the rate of template switching and Ura⁻ frequencies. Please include the original location used in figure 1 and 2 on figure 3b and 3c.

7. Figure 3d: in the 35 kb AO condition, I would expect to see more ND or ura mutations given your data in Figure 1d and the trends for 75 kb, 140 kb and ch. II. Given the increased frequency of Ura⁻ colonies (figure 3c), it is possible that the current data does not reflect the true distribution of outcomes and more colonies should be tested.

8. Orthologs of fml1 Δ and pfh1 Δ also have opposing roles in break induced replication (BIR). Loss of MPH1 (FML1 ortholog in *S. cerevisiae*) increases BIR rates (10.1534/genetics.115.184317).

Conversely, PIF1 deletion (PFH1 ortholog in *S. cerevisiae*) abolishes BIR (10.1038/nature12584). Is it possible that the gene replacement is done via BIR or MIDAS like mechanisms rather than excision and TGR? Have you tested a pol32 Δ mutant?

a. Along this line, couldn't BIR be an explanation for your ND colonies (ectopic BIR) or your alternative recombinants (longer tract gene conversion).

9. Is RPA required for the formation of dup-dels in the same way that Rad52 is? Does overexpression of RPA increase dup-del formation (conversely, does impairment of RPA decrease dup-dels)?

10. Figure 6 in text: The rationale and conclusions for this are underdeveloped. Please elaborate on why this experiment was important and what the outcome means for the greater hypothesis.

11. Questions about your model:

a. Step 3: wouldn't the invasion take place at the indicated homology and not before it? Aren't there two different homologies (the TEF promoter and terminators)?

b. Step 4: Does the converging fork result in disengagement of the elongating strand (word choice issue with whilst)?

c. Step 6: Is it possible to distinguish between your two models (Figure 6e, supplementary figure 9) by looking at nuclease requirements? Mus81 or the Slx4 scaffold would be important in resolving the 4-way junction in model figure 6e whereas flap endonucleases like Rad27 (Fen1) or Pso2 would be required in your model in supplementary figure 9.

Minor Concerns:

1. Figure 1e, 1f: all your data presents IO first and then AO. It would be helpful to also present these figures in this order.

2. Line 129-132: This is an awkward way to present the data because you're discussing two different experimental conditions. You should present the AO and IO data separately and not combine them.

3. Line 148-149: Earlier in your paper you define upstream and downstream in relation to the RFB. The "rightward" moving fork is confusing terminology. Please switch to converging fork or downstream fork.

4. Figure 6: rename the dup-del outcomes from Dup-Del1 and Dup-Del3a. This is a separate assay system and shouldn't be confused with the assay in figure 1a.

We thank the reviewers for their insightful comments and suggestions, and hope that our revised manuscript meets with their approval. Our responses to their points are in blue text.

Reviewer #1 (Remarks to the Author):

The authors use a genetic reporter assay to study the effect of a site-specific replication fork block on the frequency of both RDR (recombination-dependent replication) events and duplication-deletion (Dup-Del) rearrangements in fission yeast. They show that a site-specific RTS1 barrier can induce high frequency Dup-Del rearrangements. The Dup-Del events described use short sequence homologies flanking the HPH drug resistance cassette inserted next to the RTS1 site to replace a *ura4* cassette flanked by the same homologies on the unreplicated side of the fork barrier. The authors claim that these events occur through a model they call DRFT (Duplication by Reversed Fork Termination), a mechanism they state is similar to targeted gene replacement (TGR) based on the genetic requirements, and are independent from RDR or Rad51-dependent multi-strand invasion. In this model, they propose that fork stalling can lead to the over-replication of DNA at a reversed fork upon convergence with an oncoming replication fork, as well as the eventual release of a DNA fragment that can then be integrated at another site in the genome using the short flanking homologies.

The data reported are intriguing but I am not completely convinced that the Dup-Del events occur by the mechanism proposed for the following reasons:

1. The data presented in Figure 2 is not consistent with the need for a converging fork. Surely if the arrival of the converging fork were to be delayed (*ori1253* deletion), it would reduce the frequency of Dup-Del events? Could another RTS1 barrier be inserted to interfere with replication initiated at centromere proximal sites to reduce fork convergence to directly test this aspect of the model?

Our response:

The timing of replication fork encounter with *RTS1* and subsequent fork convergence varies in each cell cycle depending on which telomere and centromere proximal origins fire. As discussed previously, this creates a variable window of time for Rad52 to be recruited to the stalled/collapsed fork (see Nguyen et al 2015 PMID: 25806683). When *ori-1253* is present, this window is not long enough in every cell to allow a detectable Rad52 focus to form at the barrier (Nguyen et al 2015). Deleting *ori-1253* increases the time between first fork encounter with *RTS1* and fork convergence such that almost all cells now exhibit a Rad52 focus that colocalizes with the barrier (Nguyen et al 2015). However, whilst in some cells there is now sufficient time for RDR to progress downstream of the barrier, in other cells there is not. Therefore, we think that reductions in Dup-Del formation that result from fork convergence occurring at sites downstream of the barrier are likely offset by the overall increase in the number of cells in which fork reversal and Rad52 recruitment occurs prior to fork convergence.

Importantly, we do not see an increase in Dup-Del formation upon *ori-1253* deletion, which is in stark contrast with the observed increase in direct repeat recombination at the barrier (both gene conversions and deletions) and RDR-associated template

switching downstream of the barrier (Figure 2) (Nguyen et al 2015). This result indicates that Dup-Del formation does not simply depend on the recruitment of recombination proteins to the stalled fork, and strongly suggests that fork convergence is also important.

We have added the following text to the Discussion to explain the importance of the *ori-1253Δ* data:

*“A key finding that suggests that fork convergence is necessary for Dup-Del formation is the failure of *ori1253* deletion to increase Dup-Del frequency. From previous work, we know that delaying the oncoming replication fork by deleting *ori1253* results in an overall increase in recombination at *RTS1* as determined by: 1) a greater proportion of cells exhibiting a *Rad52* focus at the barrier; 2) a corresponding doubling of recombination between direct repeats flanking the barrier; and 3) an increase in RDR-associated template switching downstream of the barrier.^{17, 22} Despite this overall increase in recombination activity, we see no increase in Dup-Del formation. This implies that Dup-Del formation does not simply depend on the recruitment of *Rad52* to the *RTS1* barrier and the general promotion of recombination – it indicates that another factor is required, which is most likely fork convergence. Deleting *ori1253* extends the window of time between first fork encounter with *RTS1* and fork convergence in the majority of cells.¹⁷ However, the duration of this window will vary from cell to cell depending on which of the telomere proximal origins fire so, whilst in some cells there will be sufficient time for RDR to initiate and progress downstream of the barrier, in other cells it will only be sufficient to enable recruitment of *Rad52*. Therefore, we think that reductions in Dup-Del formation, that would result from fork convergence at sites downstream of *RTS1*, are offset by the overall increase in the number of cells in which fork reversal and *Rad52* recruitment occurs prior to fork convergence. This explains why deleting *ori1253* does not result in a reduction in Dup-Del frequency.”*

We agree that creating a condition where all collapsed forks are restarted and fork convergence at the *RTS1* barrier is prevented would be a good test for the fork convergence aspect of our model. However, we do not think that placing a second copy of *RTS1* at a centromere proximal site is the best way to achieve this as it would likely result in recombination between the two *RTS1* sites (see Lambert et al, 2005 PMID: 15935756; Mizuno et al 2009 PMID: 20008937). As there are a number of essential genes in this genomic region, this inter-*RTS1* recombination would most likely make the strain too unstable to work with. We are working on developing strains where we can more effectively restrict fork convergence at *RTS1*. However, our finding that the DNA region downstream of the replication fork barrier can form part of the duplicated region that makes up the Dup-Del (see Figure 7), suggests that Dup-Del formation can occur after replication has restarted. As depicted in our model (Figure 8), the instability of RDR, which underlies its high frequency of template switching, means that the oncoming replication fork may be able to pass the DNA region extended by RDR resulting in longer fragments of over-replicated DNA. Therefore, simply delaying the oncoming replication fork may never eradicate fork convergence at the *RTS1* barrier and the Dup-Del formation that can result from it.

2. The frequency of Dup-Del events is dramatically reduced as the *ura4* cassette is

placed further from the RTS1 barrier and undetectable when it is on another chromosome. This distance effect suggests a cis interaction and is not consistent with release of the fragment from the stalled fork. Why does the fragment need to be released? It seems just as plausible for the resected reversed fork to invade the *ura4* cassette and cleavage to be associated with strand assimilation. The authors cite work from the Ira lab showing increased insertions at a targeted DSB in the budding yeast *dna2* mutant as support for their model. However, in that study the insertions were in trans not in cis. Moreover, the authors did not test the *dna2* conditional mutant to see if the frequency of Dup-Del events increases.

Our response:

The Ira lab study did not measure whether templated insertions were more common when replication fork stalling was closer to the DSB. The study simply recorded that templated insertions were detectable in a *dna2* mutant background.

Whilst the frequency of Dup-Dels does decline with distance from the *RTS1* barrier, Dup-Del events are detectable when the *ura4* cassette is placed on another chromosome (see Figure 3d). Therefore, it is clear that *RTS1*-induced templated insertions can occur in trans and, in this regard, our findings are similar to those of the Ira lab.

We think that nearby (cis) templated insertions are more frequent because the diffusion of the over-replicated DNA fragment within the nucleus is constrained. This will restrict the opportunity for it to come into contact with more “distant” genomic sites before it has been degraded.

We have added the following to the Discussion section to highlight the “distance effect” and to explain how the movement of the DNA fragment may be constrained.

“Whilst we have detected inter-chromosomal Dup-Del events, Dup-Del frequency does decline with genomic distance between donor and recipient sites. Several factors could contribute to this distance effect including: 1) the half-life of the over-replicated DNA; 2) chromosomal organisation within the nucleus⁵²; 3) anchorage of the collapsed replication fork at the nuclear pore complex⁵³; 4) the compartmentalisation of the over-replicated DNA into a nuclear condensate^{54, 55}; 5) the apparent immobility of longer DNA fragments in the nucleoplasm^{56, 57}; 6) the possibility that the over-replicated DNA is assimilated at the ectopic site prior to its release from the site of fork convergence (Version 2 of the DRFT model). Even with the proposed protection by Rad52, we imagine that the over-replicated DNA will eventually be degraded and, therefore, has a finite time in which to encounter a genomic site where it can be integrated. Therefore, if the movement of the DNA fragment within the nucleoplasm is constrained, then nearby sites will be the ones most frequently targeted. In future studies it will be important to determine whether there is a dedicated pathway for limiting the genomic re-insertion of the over-replicated DNA via its containment and degradation.”

We agree with the reviewer that we cannot discount the possibility that the excess DNA is assimilated at an ectopic site before it has been detached from the site of fork stalling. We have revised our model (see Figure 8) to include this possibility as one of three variants of DRFT.

We have not tested whether Dna2 suppresses Dup-Del events as it is an essential gene and, unlike in budding yeast, a suppressor has not been identified in fission yeast to enable this experiment (this is something we are working to address).

3. In support of their model, loss of Rad52 strand annealing activity, Rad16 (Sc Rad1), and Msh2 all decrease the frequency of Dup-Del formation, and all of these genes are also required for TGR in budding yeast. To my knowledge these mutants have not been shown to be defective for TGR in fission yeast. The increase in Dup-Del events in the absence of Rad51 is unexpected since TGR is decreased in HR-defective mutants. It would be of interest to determine whether TGR is Rad51-independent in fission yeast and has the same dependency on Rad16. This could be easily tested by amplifying the HPH cassette with flanking homologies and transforming the fragment (as ssDNA or dsDNA) into cells to replace the *ura4* cassette (preferably in a strain with no HPH cassette).

Our response:

We are currently carrying out a detailed analysis of the genetic requirements of TGR in fission yeast that will be reported in a future paper. In the meantime, we have modified our text to make clear that our assumptions about TGR are based on work done in budding yeast rather than fission yeast. We have also included the following text to explain how a reduction in TGR in a *rad51*Δ mutant could be offset by an increase in the formation of the over-replicated DNA.

"Altogether these data show that RTS1-AO induced Dup-Del formation has similar genetic requirements as TGR in S. cerevisiae, which is consistent with our proposed DRFT model. One exception is Rad51 that, despite being required for TGR in S. cerevisiae, suppresses Dup-Del formation. Interestingly, whilst the frequency of TGR in S. cerevisiae is reduced by ~1000-fold in a rad52Δ mutant, it is only reduced by ~8-fold in a rad51Δ mutant indicating that TGR is not completely dependent on Rad51.³⁰ We suspect that the increase in Dup-Dels observed in a rad51Δ mutant reflects an inhibitory effect of Rad51 prior to TGR. We also think that this increase compensates for any reduction in TGR that a rad51Δ mutant might exhibit."

4. If resection is important, would an *exo1*Δ *rqh1*Δ be more definitive compared to using either of the single mutants?

Our response:

We have added a new Results section, which includes the *exo1*Δ *rqh1*Δ double mutant data (see new Figure 5). We show that the frequency of Dup-Dels is reduced by ~4-fold in the double mutant suggesting that long-range DNA end resection is required for efficient Dup-Del formation.

5. Finally, the DRFT model predicts fork reversal and cleavage of the regressed arm. However, of the helicase mutants tested, only Pfh1 reduced the frequency of Dup-Del events, and there is the complication of the role of Pfh1 in termination. Are there other translocases that function in fork reversal in fission yeast that could be tested? Although Dup-Del events are reduced in the *rad16* mutant, there is no evidence that Rad16 cleaves a reversed fork and it seems more likely that it is acting at the

heterology boundary during integration of the fragment. Mus81 has been shown to act at stalled and reversed forks and should also be tested in this assay.

Our response:

As noted by the Reviewer, we tested several DNA helicases/translocases that could potentially drive fork reversal (Fbh1, Fml1, Pfh1, Rqh1 and Srs2). As reported in our paper, only Pfh1 was required for Dup-Del formation. We are currently developing a high throughput forward genetic screen that will allow us to identify other pro-Dup-Del factors, which will ultimately enable us to identify the fork reverser(s).

As reported in our paper, Rad16-Swi10 strongly promotes Dup-Del formation and, whilst there is no evidence that it cleaves reversed replication forks, it could liberate the over-replicated DNA if it is resolved into a 3' flap during fork convergence (see Version 3 of our DRFT model in Figure 8).

We have tested *mus81Δ* and have included this data in Supplementary Figure 8 - it has no effect on Dup-Del frequency.

Other comments:

6. Plotting Ura- frequency: In Figure 3C, the data are fitted to a log scale. This seems to be a much better visual representation of the data (although I'm not sure why a few of the data sets in 3C don't have bars) than using a linear scale. In 1C, 2C, 4A, 5A, and 6B, the reader can't appreciate the spread of the data since many of the data sets hug the x-axis. This would be an easy fix and improve the figures readability significantly.

Our response:

For the control (*RTS1-IO*) strains many of the colonies assayed produced zero FOA resistant colonies and, as these data won't show on a log scale plot, we used a linear scale for most graphs so that all the data points can be seen. This explains why the data for the *RTS1-IO* strains tends to hug the X-axis (i.e. individual data points are either very low or zero). It also explains why some of the *RTS1-IO* data sets in Figure 3c don't have error bars as the median values are zero and the 95% confidence intervals are below the 0.001 mark on the Y-axis. However, all of the data are shown in Supplementary Table 1, and we have included a Source Data file that shows all the colony counts, calculated frequencies and results of the colony PCR analysis.

7. Inclusion of IO and AO data: While I understand the rationale for inclusion of both IO and AO data in the earlier figures to validate that the reporter works and the RTS1 block is directional, I don't see the need to include these data in Figures 4 and 5. Given that these figures are probing the genetic requirements of dup-del formation in response to replication fork stalling, it seems that the IO data doesn't contribute to the model. These data could probably be moved to the supplement. In that case, Figures 4 and 5 could be combined into one four panel figure. Alternatively/additionally, this would free up space for discussion of other mutants such as *exo1Δ* to be presented in the main text or to use Figure 6 only to describe different models for Dup-Del generation.

Our response:

We think that it is important to include the *RTS1*-IO data in Figures 4 and 5 (now Figure 6) because it allows the reader to see which mutants retain residual *RTS1*-AO-induced Dup-Dels. For example, it allows the reader to see that the frequency of Dup-Dels in a *pfh1-mt** mutant is still higher than the spontaneous level without having to resort to the data in Supplementary Table 1. However, we have split Figure 6 into two figures (new Figure 7 and 8), so that the different models for Dup-Del generation can be displayed together in one figure. We have also moved the *exo1Δ* data to the main text (new Figure 5).

8. Presenting breakdown for % of Ura⁻ colonies: The n values for colonies scored for Dup-Dels or *ura* mutations vary significantly. While I don't think all strains tested need to have dozens of colonies scored, certainly an n value of 2 is not representative of all the events detected in these assays. There should be some minimum threshold for how many colonies are scored.

Our response:

As mentioned above, many of the colonies assayed for the *RTS1*-IO strains contained zero FOA resistant cells (because spontaneous Dup-Dels are a very rare event). For example, in Figure 3d the *RTS1*-IO strains with the *ura4* cassette at 75 kb and 140 kb downstream of *RTS1* produced FOA resistant colonies from only 2 out of 15 and 3 out of 20, respectively, of the initial colonies assayed (see Source Data File). For most assays, we only assayed one FOA resistant colony for each initial colony that produced FOA resistant colonies to avoid analysing clonal populations (hence the n values of 2 and 3 in Figure 3d). The one exception is for the strains with the *ura4* cassette on chromosome 2, where all FOA resistant colonies were assayed (see Source Data File). We have modified the Methods section to make these points clearer.

9. Figure 1A, 2A, 3A, 6A: The reporter used in these experiments as currently presented is a bit overwhelming. For example, I'm unsure as to why so many of the origins are labeled. It would be a lot simpler to only label the origins suspected to be firing through the reporter. The figure could also be simplified by removing the arrows indicating direction of transcription of the various parts of the reported since transcription is not mentioned in the text. Finally, it might be nice to have *RTS1* as a stop sign or triangle to indicate that it's not simply part of the reporter; it is initiating the fork stall (as is done in the model).

Our response:

We have decluttered the schematics and included a stop sign for *RTS1* as requested. However, we have retained the replication origin labels, and arrows indicating the direction of transcription, as we think this information is useful.

10. The recombination events in this assay seem to be extremely prevalent (1 in 1,000 cells). I find this somewhat surprising given the mechanism proposed. Can the authors comment on this?

Our response:

We think this is due to the fact that fork stalling at *RTS1*, followed by recruitment of Rad52, happens in the majority of cells in each cell cycle (see Nguyen et al 2015).

Therefore, the over-replication problem will be happening in most cells in every generation creating ample opportunity for the excess DNA to be assimilated at an ectopic site. Whilst the frequency of the Dup-Del events is ~1 in 1000 cells, this is still much lower than the frequency of template switching which is ~1 in 10 when *ori-1253* is deleted.

Reviewer #2 (Remarks to the Author):

The new manuscript by Oehler et al. presents a new mechanism leading to deletion/duplication. They use a fission yeast system to study genome instability. The experimental model has reporter markers URA4 and *hygR* and replication fork barrier sequence (*RTS1*) in between the two markers. *RTS1* is positioned in such a way that replication forks encountering *RTS1* can produce reversed forks within the *hygR* sequence. *hygR* sequence is flanked on both ends by 250-400 bp sequences homologous to URA4 flanking sequences and other upstream sequences and therefore can lead to recombination generating *ura* minus colonies that can be scored on FOA plates. Fork blocking induced such dup-del events by nearly 2000 times. When the URA4 marker was further away or even on different chromosomes dup-del recombination was also induced by fork blocking, however at lower frequencies. Recombination was also not reduced but rather increased in the absence of Rad51 excluding regular template switching mechanism. Genetic analysis shows the requirement for Rad52 annealing activity, Rad1-Rad10 nuclease, and Msh2. This is an identical gene set of genes needed for gene targeting suggesting a model where duplicated region at the reversed fork may be released and integrated elsewhere in the genome.

The strengths of this interesting work are:

Demonstration that replication fork barrier/reverse replication forks can lead to duplication-deletion and comprehensive genetic analysis of these events suggesting possible mechanism.

The manuscript is well written.

Comments

1. Could sister chromatid recombination with no crossover contribute to URA4 X *hygR* recombination? Fasullo's lab demonstrated that spontaneous inter-sister recombination could occur in *rad51* minus mutant in budding yeast.

Our response:

In principle, Dup-Dels could be formed if the donor site invades the recipient site prior to fork convergence. However, if such a mechanism operates, we would expect to see an increase in Dup-Del formation when *ori-1253* is deleted. Simply increasing overall recombination activity, by extending the window of time between first fork encounter with *RTS1* and the arrival of the oncoming fork, does not increase Dup-Del frequency (see our response to Reviewer 1's first point).

2. Could strand assimilation occur when over-replicated region is not yet liberated? This could explain the higher rate of dup-del with a reporter gene in proximity to

RTS. Alternatively could local supercoiling change with a reversed fork that could open duplex DNA and facilitate annealing?

Our response:

We cannot exclude the possibility that the over-replicated DNA is assimilated at the ectopic site prior to its liberation from the donor site. This possibility is included in the revised version of our Discussion and Figure 8 (please see our response to Reviewer 1's second point).

The finding that *ori-1253Δ* does not increase Dup-Dels, suggests that fork reversal and recruitment of Rad52 are insufficient to drive Dup-Del formation (see our response to point 1 above). This means that any changes in local supercoiling that might be caused by fork reversal are also insufficient on their own to drive Dup-Del formation.

3. It is not tested/discussed which nuclease could be involved in the release of a duplicated region. In discussion, Rad1/Rad10 is suggested.

Our response:

We have included additional data showing that Mus81 is not required for Dup-Del formation and modified our Discussion to highlight the potential role of Rad16-Swi10 (Rad1-Rad10) in liberating the over-replicated DNA in the third variant of our DRFT model (Fig. 8).

4. Please provide exact homology between all H sequences flanking markers and others. How many mismatches are present?

Our response:

We have included this information in the Supplementary Figure 3 legend.

Reviewer #3 (Remarks to the Author):

The submitted study by Oehler, Morrow and Whitby utilize the well-studied RFB system, and a marker based genetic assay in *S. pombe* to investigate mechanisms that drive gene duplication events. These duplication events occur during DNA replication fork stalling and fork reversal, which the authors term duplication by reversed fork termination (DRFT) wherein a canonical replication fork merges with a stalled reversed fork resulting in the over-replication of DNA. The excess DNA, if not degraded, can then be integrated at an ectopic site via a mechanism like targeted gene replacement (TGR) and result in duplication-deletion events (dup-dels). The authors identify that the formation of these dup-dels requires Rad52, Rad16-Swi10, and Msh2. They show loss of Ku70 and Rad51 increase dup-del formation and hypothesize that their absence could result in impaired fork protection. Additionally, the helicases Fml1 and Phf1 have opposing roles in dup-del formation due to their roles in fork restoration and fork merging, respectively. Altogether, the authors suggest that DRFT could contribute to genomic instability and structural variations in the genome, including those associated with cancer genomes.

While I believe this study's conclusions are supported by compelling data and

presents an interesting mechanism for genomic instability and structural variation, there is room for improvement in the text and scientifically. The manuscript would be much improved with more details throughout. Of note, much of this manuscript is lacking adequate context to set up the rationale for experiments (Figure 2 and Figure 6 in particular). It would also help if the findings throughout were concluded in ways that linked back to the initiating rationale. Additionally, there are significant inconsistencies among presented data which need to be clarified before it would be suitable for publication.

Major Concerns:

1. In the manuscript, you don't discuss the possible mechanisms by which you get no amplification by PCR (ND). Have you done Southern blotting to look at the restriction fragments at your loci? Discussion and determination of this population is essential because it is not a small population (many have 20% or more ND).
a. Along these lines, you mention in line 129-130, of the 40 *ura*⁻ colonies you tested that are in the AO condition all of them are Dup-Dels but your graph in 1d suggests that 20%-30% should be colonies where no detectable PCR occurs. Please explain the discrepancy between your data in 1d and figure 1e, f, Sup Figure 2 and 3.

Our response:

The vast majority of NDs arise because of a technical failure of the colony PCR. In particular colony PCRs performed with primer set B had a relatively high failure rate despite working 100% of the time when purified genomic DNA was used. Because in most cases it was only the PCR with primer set B that failed, the majority (~77%) of the ND results do provide some useful information as they show whether *ura4* has been retained or replaced by Hyg (this information is presented in the Source Data file).

We have added the following text to explain the NDs and the apparent discrepancy between the data in Figure 1d and Figures 1e, f and Supplementary Figures 2 and 3:

"Unlike the colony PCR analysis, where some PCRs failed to generate a DNA band, all of the PCRs done with purified genomic DNA yielded a band. This suggests that cases of undefined Dup-Del status by colony PCR analysis arise mainly because of a technical problem with the PCR rather than some undetermined genomic rearrangement that cannot be amplified with primer sets A or B."

We have also included the following text in the Methods to direct the interested reader to the Source Data file:

*"A minority of the colony PCRs failed to yield a DNA band and, therefore, the Dup-Del status of these colonies remained undetermined (indicated by ND in the relevant Figures). However, in approximately 77% of all ND cases only the PCR for primer set B failed and, therefore, we could determine whether the *ura4* gene had been retained or replaced by *hyg*^R. Approximately 83% of these colonies were the latter type where the *ura4* gene had been replaced by *hyg*^R. This information is recorded in the Source Data file, which includes the raw data for the recombination assays and all PCR analysis."*

2. The data in figure 1d presented in the paper a few times but is not consistent— Figure 2d has the same test population (RTS-AO + origin) and it has a completely different distribution of outcomes. The data seems to match in figure 1d and 4b. This is important because claims related to figure 2d state there is no difference between the presence or absence of the origin of replication, but if the data in 2d is to be believed then loss of the origin increases ND and eliminates mutation or URA3. Please clarify your graphs and the conclusions you can draw from these data.

Our response:

Thank you for spotting this mistake. When drawing Figure 2d we had inadvertently transposed the two middle bars. This mistake has been corrected in the revised version of our paper.

3. In figure 2b and 2c you are measuring two separate phenomena: RDR-associated template switching and dup-dels. The incidence of ADE+ reversion is very high (10⁻¹ or 10⁻²). What is the co-incidence of *ura-* *ade+* cells, which would be colonies where dup-dels and template switching resulted in mutational repair? Are *Ura-* colonies more likely to be *Ade+*?

Our response:

We have determined the co-incidence of *Ura-* *Ade+* cells (see new Supplementary Table 2). The frequency of *Ade+* cells amongst *Ura-* cells is the same as in the total population, which indicates that RDR-associated template switching and Dup-Del formation are independent processes.

4. Previous work from the Carr lab (10.1038/s41467-021-21198-0) showed that RDR is completed by a δ/δ fork on the leading and lagging strand and that less processive synthesis leads to increased gaps (which could a mechanism for microsatellite mediated instability: (10.1371/journal.pgen.1009863). Linking point 3 and 4, could remaining post-replicative gaps be sites of annealing between the cleaved reversed fork and the region of homology that will be deleted?

Our response:

We think it is unlikely that post-replicative gaps caused by RDR are required for Dup-Del formation as we detect no increase in Dup-Del formation in an *ori-1253 Δ* background, where RDR-associated post-replicative gaps at the *ura4* site would be more prevalent. Moreover, we find no increase in the frequency of *Ade+* recombinants amongst Dup-Del recombinants compared to non-Dup-Del colonies. This suggests that Dup-Del formation is not linked to RDR (see our response to point 3 above).

a. Ln 142-153 does a poor job describing the combinatorial assay (*ADE+* reversion and *ura-*), the previous work and what your data output means in terms of your developing model. Please add additional context to the description of the assay and consider adding a replication fork model demonstrating the possible outcomes.

Our response:

We have extended the text as requested (please see lines 161 – 199 in the revised manuscript).

5. The increase in dup-del rate in the *rad51Δ* mutant is surprising and your hypothesis about Ku and Rad51 is intriguing (In 215-216). What happens in the *rad51Δ kuΔ* double mutant in terms of dup-dels?

Our response:

Whilst analysis of a *rad51Δ ku70Δ* double mutant would be interesting it is not integral to our current paper. We are currently working through a list of mutants to further define the genetics of Dup-Del formation, and this work will form part of a future paper.

6. It's difficult to ascertain whether movement to 35 kb changes the rate of template switching and Ura⁻ frequencies. Please Include the original location used in figure 1 and 2 on figure 3b and 3c.

Our response:

Done

7. Figure 3d: in the 35 kb AO condition, I would expect to see more ND or ura mutations given your data in Figure 1d and the trends for 75 kb, 140 kb and ch. II. Given the increased frequency of Ura⁻ colonies (figure 3c), it is possible that the current data does not reflect the true distribution of outcomes and more colonies should be tested.

Our response:

As explained above (see our response to point 1), the NDs mainly reflect colony PCR reactions that failed due to technical reasons. Therefore, the only real difference between the 11.4 kb and 35 kb sites is that the frequency of Dup-Dels drops by ~3-fold at the latter site.

8. Orthologs of *fml1Δ* and *pfh1Δ* also have opposing roles in break induced replication (BIR). Loss of MPH1 (FML1 ortholog in *S. cerevisiae*) increases BIR rates (10.1534/genetics.115.184317). Conversely, PIF1 deletion (PFH1 ortholog in *S. cerevisiae*) abolishes BIR (10.1038/nature12584). Is it possible that the gene replacement is done via BIR or MIDAS like mechanisms rather than excision and TGR? Have you tested a *pol32Δ* mutant?

a. Along this line, couldn't BIR be an explanation for your ND colonies (ectopic BIR) or your alternative recombinants (longer tract gene conversion).

Our response:

BIR is restricted by fork convergence in the same way as RDR (see Mayle et al 2015). Therefore, if BIR was responsible for Dup-Del formation, we would expect to see an increase in Dup-Del frequency when *ori-1253* is deleted. Also, BIR exhibits a greater dependence on Rad51 than RDR and, therefore, the increase in Dup-Del frequency that we see in a *rad51Δ* mutant is not consistent with them arising by a BIR-like mechanism. We have added new data to our revised manuscript showing that Mus81 is not required for Dup-Del formation (see Supplementary Fig. 8). As MiDAS requires Mus81 (Bhowmick et al 2016), our new data suggest that Dup-Del formation does not depend on a MiDAS-like mechanism.

The Pol32 ortholog in fission yeast (*Cdc27*) is essential and, therefore, we haven't been able to test whether it is required for Dup-Del formation.

9. Is RPA required for the formation of dup-dels in the same way that Rad52 is? Does overexpression of RPA increase dup-del formation (conversely, does impairment of RPA decrease dup-dels)?

Our response:

Whilst these are interesting questions, they are not integral to our current paper and, therefore, we will seek to address them in future work.

10. Figure 6 in text: The rationale and conclusions for this are underdeveloped. Please elaborate on why this experiment was important and what the outcome means for the greater hypothesis.

Our response:

Done (see lines 367-387 in the revised manuscript).

11. Questions about your model:

- a. Step 3: wouldn't the invasion take place at the indicated homology and not before it? Aren't there two different homologies (the TEF promoter and terminators)?
- b. Step 4: Does the converging fork result in disengagement of the elongating strand (word choice issue with whilst)?
- c. Step 6: Is it possible to distinguish between your two models (Figure 6e, supplementary figure 9) by looking at nuclease requirements? Mus81 or the Slx4 scaffold would be important in resolving the 4-way junction in model figure 6e whereas flap endonucleases like Rad27 (*Fen1*) or Pso2 would be required in your model in supplementary figure 9.

Our response:

a. Our model depicts invasion of the DNA strand at the reversed fork into the homologous DNA (immediately ahead of the reversed fork) followed by extension of the invading strand beyond the *RTS1* barrier by RDR involving a migrating D-loop. We have added an extra panel and additional explanatory text to our model (Figure 8) to make this clearer.

b. In our model, the migrating D-loop is unstable meaning that the DNA strand that is being elongated by RDR can dissociate from its template. It is during these moments of strand disengagement, that we think an oncoming canonical replication fork can move past the elongated strand and converge with the reversed fork. This point is now made clearer in the revised version of our model (Figure 8). Whether the oncoming replication fork can drive the dissociation of the strand that is being elongated by RDR, rather than merge with it, remains to be determined.

c. We have tested Mus81 and found that it is not required for Dup-Del formation (these data have been added to Supplementary Figure 8 and are mentioned in the Discussion). We plan to screen all candidate nucleases as part of a follow-up study. However, our finding that Rad16-Swi10 (*Rad1-Rad10/XPF-ERCC1*) is required for Dup-Del formation, suggests that this nuclease could be responsible for liberating

the over-replicated DNA if it is converted into a 3' flap during fork convergence (see Figure 8 and accompanying Discussion).

Minor Concerns:

1. Figure 1e, 1f: all your data presents IO first and then AO. It would be helpful to also present these figures in this order.

Our response:

Done.

2. Line 129-132: This is an awkward way to present the data because you're discussing two different experimental conditions. You should present the AO and IO data separately and not combine them.

Our response:

Done.

3. Line 148-149: Earlier in your paper you define upstream and downstream in relation to the RFB. The "rightward" moving fork is confusing terminology. Please switch to converging fork or downstream fork.

Our response:

The term "rightward" has been removed from the revised text.

4. Figure 6: rename the dup-del outcomes from Dup-Del1 and Dup-Del3a. This is a separate assay system and shouldn't be confused with the assay in figure 1a.

Our response:

We have renamed them Dup-Del 1.1 and Dup-Del 3.1

REVIEWERS' COMMENTS

Reviewer #1 (Remarks to the Author):

I appreciate the authors thoughtful written responses to my comments, but they have not addressed many of the concerns experimentally. Other than Rad16-Swi10, no other nuclease tested seems to impact DRFT. I realize that Dna2 is essential, but don't see why the authors could not use the ts allele that has been used in other studies in *S. pombe* (see Kang et al., Genetics 2000).

The new Figure 8 is an improvement, and versions 2 and 3 seem more likely as the genetic requirements are consistent with single-strand assimilation and I am not aware of a nuclease that could cleave the reversed fork as shown in version 1. The decrease in Dup-Dels observed in the *exo1 rqh1* double mutant is consistent with the need for resection of the reversed fork to generate a long 3' tail, but contrasts with the large increase in TGR observed in the absence of long-range resection in budding yeast. Some discussion of this point is needed.

Reviewer #2 (Remarks to the Author):

The revised manuscript is improved, questions of this reviewer were addressed. Most importantly, additional possibilities are presented with respect to possible mechanism of gene duplication.

Reviewer #3 (Remarks to the Author):

The authors have addressed my concerns and clarified their paper sufficiently. I also understand some reviewer suggestions are the topic of future work and I agree that this paper presented is a complete story. Overall, the revised manuscript improved in text and experimentally and I would recommend this paper for publication.

We thank the reviewers for their additional comments and suggestions, and hope that that the final version of our manuscript meets with their approval. Our responses to their points are in blue text.

Reviewer #1 (Remarks to the Author):

I appreciate the authors thoughtful written responses to my comments, but they have not addressed many of the concerns experimentally. Other than Rad16-Swi10, no other nuclease tested seems to impact DRFT. I realize that Dna2 is essential, but don't see why the authors could not use the *ts* allele that has been used in other studies in *S. pombe* (see Kang et al., Genetics 2000).

The new Figure 8 is an improvement, and versions 2 and 3 seem more likely as the genetic requirements are consistent with single-strand assimilation and I am not aware of a nuclease that could cleave the reversed fork as shown in version 1. The decrease in Dup-Dels observed in the *exo1 rqh1* double mutant is consistent with the need for resection of the reversed fork to generate a long 3' tail, but contrasts with the large increase in TGR observed in the absence of long-range resection in budding yeast. Some discussion of this point is needed.

Our response:

As our genetic assay requires that the cells remain viable, we would be unable to use the *dna2 ts* allele at its restrictive temperature. However, in future work, we will investigate whether the *dna2 ts* allele has any effect on Dup-Del formation at semi-permissive temperatures.

We have included the following about the *exo1Δ rqh1Δ* double mutant in the Discussion:

*"We also found that a *rqh1Δ exo1Δ* double mutant was deficient in Dup-Del formation, suggesting that long-range DNA end resection promotes DRFT. In budding yeast, the deletion of the RecQ-type helicase Sgs1, together with Exo1, has been shown to either increase or decrease the efficiency of TGR depending on the assay used.^{62, 63} Therefore, the reduction in Dup-Del formation that we observe in a *rqh1Δ exo1Δ* double mutant could result from less efficient TGR. Alternatively, if Rqh1 and Exo1 suppress TGR, then their role in promoting long-range DNA end resection at the stalled replication fork presumably supersedes this effect. In future studies, it will be important to distinguish between effects on the production of the over-replicated DNA and its subsequent integration into a new chromosomal site."*

Reviewer #2 (Remarks to the Author):

The revised manuscript is improved, questions of this reviewer were addressed. Most importantly, additional possibilities are presented with respect to possible mechanism of gene duplication.

Our response:

We thank the Reviewer for their approval.

Reviewer #3 (Remarks to the Author):

The authors have addressed my concerns and clarified their paper sufficiently. I also understand some reviewer suggestions are the topic of future work and I agree that this paper presented is a complete story. Overall, the revised manuscript improved in text and experimentally and I would recommend this paper for publication.

Our response:

We thank the Reviewer for their approval.